# MPCACHE: MPC-FRIENDLY KV CACHE EVICTION FOR EFFICIENT PRIVATE LLM INFERENCE

## ABSTRACT

Private LLM inference based on multi-party computation (MPC) offers cryptographically-secure protection for both user prompt and proprietary model weights. However, it suffers from large latency overhead for long input sequences. While key-value (KV) cache eviction algorithms have been proposed to reduce the computation and memory cost for plaintext inference, they are not designed for MPC and may even introduce more overhead. In this paper, we propose an accurate and MPC-friendly KV cache eviction framework, dubbed MPCache. MPCache is built on the observation that historical tokens in a long sequence may have different effects on the downstream decoding. Hence, MPCache combines a look-once static eviction algorithm to discard unimportant tokens and a query-aware dynamic selection algorithm to further choose a small subset of tokens for attention computation. As existing dynamic selection algorithms incur too much latency, we propose a series of optimizations to drastically reduce the KV cache selection overhead, including MPC-friendly similarity approximation, hierarchical KV cache clustering, and layer-wise index sharing strategy. With extensive experiments, we demonstrate that MPCache consistently outperforms prior-art KV cache eviction baselines across different LLM generation tasks and achieves $1.8 \sim 2.01\times$ and $3.39 \sim 8.37\times$ decoding latency and communication reduction on different sequence lengths, respectively. Our anonymous code repository can be found here.

## 1 INTRODUCTION

Large language models (LLMs) have recently demonstrated remarkable ability in a wide range of applications such as document summarization (Huang et al., 2021; Narayan et al., 2018; Zhang et al., 2024a), question answering (Kočiský et al., 2018; Dasigi et al., 2021; Yang et al., 2018), and dialogue systems (Thoppilan et al., 2022; Chiang et al., 2023; Taori et al., 2023). However, LLM-based machine learning as a service (MLaaS) on the cloud has raised serious privacy concerns as the users are required to upload their prompts to the cloud, which may contain sensitive personal information. Meanwhile, the service provider is unwilling to offload the trained model to the user to protect the proprietary model weights. Secure multi-party computation (MPC)-based private inference has been proposed to address the privacy concerns (Goldreich, 1998; Mohassel & Rindal, 2018; Huang et al., 2022; Rathee et al., 2020; Gupta et al., 2023). MPC enables the users and the cloud to conduct the LLM inference jointly, but nothing else can be derived beyond the final inference results.

However, MPC-based LLM inference faces serious efficiency challenges, especially for long input sequences. We profile the decoding efficiency of GPT-2 with the Secretflow framework (Ma et al., 2023) using recent 2-party computation (2PC) (Lu et al., 2023) and 3-party computation (3PC) protocols (Dong et al., 2023). As can be observed in Figure 1(a) and (b), *attention dominates the latency and communication for both 2PC and 3PC protocols. Moreover, Softmax accounts for the majority of the overall cost, especially with an increasing sequence length.*

To reduce the cost of private LLM inference, previous works focus on developing more efficient MPC protocols (Lu et al., 2023; Dong et al., 2023; Pang et al., 2023; Hou et al., 2023), replacing non-linear activation functions with more MPC-friendly operators (Liu & Liu, 2023; Li et al., 2022; Zeng et al., 2023), or directly modifying the model architecture (Rathee et al., 2024). However,

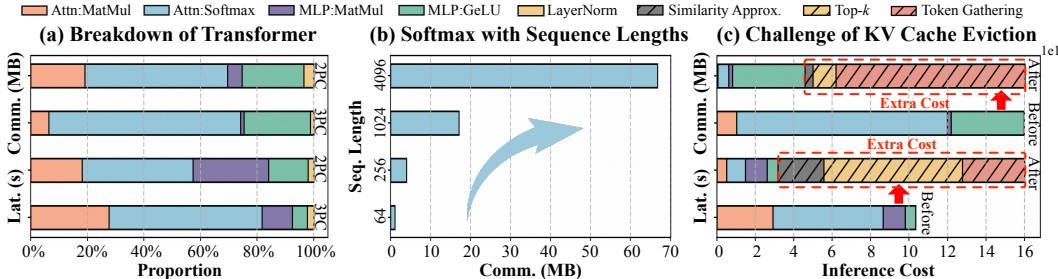

Figure 1: (a) Breakdown of decoding latency and communication for one token generation with a sequence length of 512. Attention dominates the latency and communication for both 3PC and 2PC protocols. (b) The cost of Softmax scales with the sequence length. (c) Inference cost before and after KV cache eviction. Blocks in slash indicate the extra overhead introduced by eviction.

they still incur significant overhead or require expensive finetuning or re-training, and cannot be directly applied to LLMs. Another line of works leverages key-value (KV) cache eviction to reduce the number of tokens involved in the attention computation (Zhang et al., 2024d; Ge et al., 2023; Liu et al., 2024c; Zhao et al., 2024; Zhang et al., 2024c; Fu et al., 2024). Although they have demonstrated significant memory and computation reduction for plaintext LLM inference without the need of finetuning, they are not MPC-friendly. As shown in Figure 1(c), directly applying an existing KV cache eviction algorithm (Liu et al., 2024b) incurs even more communication and latency overhead over the baseline model since it introduces expensive operators in MPC, including top-$k$ selection, token gathering, etc, as elaborated in Section 3. Therefore, there is an urgent need for an MPC-friendly KV cache eviction algorithm to improve the efficiency of private LLM inference without fine-tuning.

To overcome the heavy overhead of attention computation, we make the following observations that motivate our MPCache: 1) the LLM attention maps are overall sparse for long input prompts, motivating us to perform static eviction and directly prune the KV cache of unimportant tokens; 2) the attention maps show token-wise locality (Liu et al., 2023), motivating us to build an efficient hierarchical clustering algorithm for dynamic selection of the KV cache; 3) the attention maps of adjacent layers show similar patterns, motivating us to share the KV cache selection for adjacent layers to further improve efficiency. Our contributions can be summarized as follows:

- We observe the cost of MPC-based LLM inference mainly comes from attention computation and propose MPCache, an MPC-efficient KV cache eviction framework to reduce the LLM inference latency and communication.

- We identify the challenges when applying KV cache eviction in MPC. To tackle the problems, MPCache combines look-once static KV cache eviction and query-aware dynamic selection with a series of optimizations, including MPC-friendly similarity approximation, hierarchical KV cache clustering, and a layer-wise index sharing strategy.

- With extensive experiments, we demonstrate the performance of MPCache consistently exceeds the prior-art KV cache eviction algorithms across different generation tasks and achieves upto $2.01\times$ and $8.37\times$ decoding latency and communication, respectively.

## 2 PROBLEM FORMULATION AND BACKGROUND

### 2.1 PROBLEM FORMULATION

Generative LLM inference can be divided into prefill and generation stages (refer to Appendix A). We formally describe the generation process with KV cache eviction in Algorithm 1. The KV cache eviction policy, denoted as $\mathcal{P}$, aims to minimize the attention computation by only preserving a subset of tokens, which typically involves three steps: 1) $\mathcal{P}$ first computes the similarity between the query and key cache of previous tokens (line # 1); 2) $\mathcal{P}$ then ranks the previous tokens based on the similarity score and applies the top-$k$ algorithm to determine the indices of relevant tokens (line # 2); 3) the KV cache is then retrieved based on the indices, denoted as token gathering (line # 3),

---

**Algorithm 1:** Problem formulation of KV cache eviction for one layer

---

**Input** : Query, key, and value cache $\mathbf{q} \in \mathbb{R}^{H \times 1 \times d}$, $\mathbf{K} \in \mathbb{R}^{H \times T \times d}$, and $\mathbf{V} \in \mathbb{R}^{H \times T \times d}$, where $T, H, d$ denote the sequence length, number of heads, and embedding dimension.

**Output:** Sparse attention output $\mathbf{O} \in \mathbb{R}^{H \times 1 \times d}$.

1   $\mathbf{sim} = \mathrm{SimApprox}(\mathbf{q}, \mathbf{K})$;                 ▷ Similarity approximation

2   $\mathbf{indices} = \mathrm{topk}(\mathbf{sim}, k = k)$;                ▷ Top-$k$ selection

3   $\mathbf{K}' = \mathbf{K}.\mathrm{gather}[\mathbf{indices}]$, $\mathbf{V}' = \mathbf{V}.\mathrm{gather}[\mathbf{indices}]$;      ▷ Token gathering based on indices

4   $\mathbf{O} = \mathrm{Softmax}(\mathbf{q} \cdot \mathbf{K}'^{\top} / \sqrt{d}) \cdot \mathbf{V}'$;             ▷ Sparse attention

5   **return** $\mathbf{O}$.

---

Table 1: Qualitative comparison with prior works.

| Representative Work | Method | Similarity Approximation | Top-$k$ Selection | Token Gathering | Layer-wise Optimization | MPC Efficiency | Model Performance |
|---|---|---|---|---|---|---|---|
| Li et al. (2022) | Non-linear Replacement | - | - | - | - | **Fine-tuing Required** | **Not Applied to LLM** |
| Xiao et al. (2023) | Fixed-pattern | - | - | Token-wise | - | **High** | **Low** |
| Li et al. (2024) | Static | Accumulated Attention Score | Once during Prefill | Token-wise | - | **High** | **Low** |
| Liu et al. (2024b) | Dynamic | Token-wise Cosine Similarity | Token-wise per Step | Token-wise | - | **Low** | **High** |
| MPCache (ours) | Static+Dynamic | Hierarchical Clustering, Cluster-wise Similarity | Parallelled, Cluster-wise per Step | Cluster-wise | Adjacent Layer Sharing | **High** | **High** |

followed by sparse attention computation with the selected KV cache (line # 4). To compute the similarity in line # 1, existing works have used accumulated attention score of the historical tokens (Liu et al., 2024c; Zhang et al., 2024d; Zhao et al., 2024; Yang et al., 2024; Zhang et al., 2024c) or cosine similarity (Liu et al., 2024b; Xiao et al., 2024). KV cache eviction reduces the attention computation complexity from $\mathcal{O}(Td)$ to $\mathcal{O}(kd)$, where $T, d$ denote the sequence length and embedding dimension, respectively, and $k \ll T$. However, it introduces MPC-unfriendly operations, including similarity approximation, top-$k$ selection, and token gathering, hindering its benefits in MPC-based LLM inference. Hence, the goal of our paper can be summarized as follows:

*"How can we design an MPC-friendly KV cache eviction algorithm $\mathcal{P}^*$ to minimize MPC-based LLM inference latency without sacrificing LLM performance?"*

## 2.2 BACKGROUND

**Related works.** There has been a surge in improving the efficiency of private LLM inference. Existing works focus on the protocol optimization (Pang et al., 2023; Dong et al., 2023; Lu et al., 2023; Hou et al., 2023) or directly replace non-linear functions with MPC-friendly operators (Liu & Liu, 2023; Li et al., 2022; Zeng et al., 2023; Mishra et al., 2020; Dhyani et al., 2023). However, they either still incur large overhead for long input sequences or require expensive re-training. KV cache eviction has been widely explored for plaintext inference and can be classified into 3 categories: 1) *fixed-pattern algorithms* like Xiao et al. (2023) and Beltagy et al. (2020) always keep the tokens at the same position across generation steps, lacking flexibility for different LLMs and contexts; 2) *static algorithms* like Zhang et al. (2024d); Zhao et al. (2024); Zhang et al. (2024c); Li et al. (2024); Ge et al. (2023) discard tokens based on the accumulated attention scores of historical tokens, which are efficient as the KV cache eviction is usually only conducted once but suffer from large performance degradation when the compression ratio is high; 3) *dynamic algorithms* like Xiao et al. (2024); Tang et al. (2024b); Liu et al. (2024b) compute the similarity between the query and keys for each generation step, which is more accurate but requires repetitive selection at each generation step. Different from prior works in Table 1, MPCache is a training-free framework that combines static and dynamic algorithms, and leverages hierarchical clustering with a series of MPC-friendly optimizations, achieving high efficiency and performance simultaneously. We leave a detailed review of existing works in Appendix A.

**MPC preliminaries.** MPC (Goldreich, 1998) is a cryptographic technique recently developed and leveraged to enable LLM inference while protecting the privacy of both data and model. In an MPC framework, to protect a certain tensor, it is often split into multiple secret shares and distributed across different parties involved in the computation (Lu et al., 2023; Dong et al., 2023; Mohassel & Rindal, 2018). Dedicated protocols have been developed to support LLMs' linear and non-linear

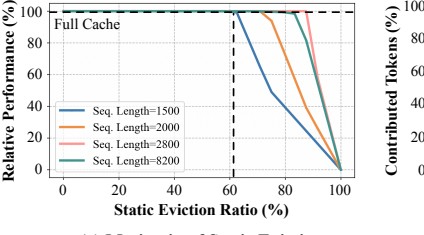 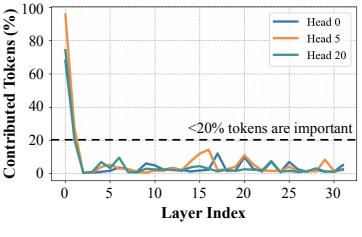 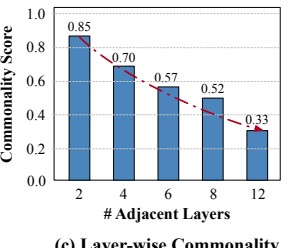

(a) Motivatin of Static Eviction  (b) Motivatin of Dynamic Selection  (c) Layer-wise Commonality

Figure 3: Motivating inspirations of MPCache. (a) Statically evicting almost 60% tokens during the prefill stage still maintains the performance; (b) less than 20% tokens contribute to token decoding; (c) layer-wise top-$k$ commonality among different numbers of adjacent layers.

operations (Lu et al., 2023; Pang et al., 2023; Dong et al., 2023). In this work, we adopt an *honest-but-curious* threat model and apply MPCache to both 2PC and 3PC protocols, which involve 2 parties and 3 parties in the computation, respectively. We refer interested readers to Appendix B, where the threat model and 2PC/3PC protocols are more clearly explained. Following Li et al. (2022); Zeng et al. (2023), MPCache is built upon existing cryptographic primitives and focuses on optimizing the LLM inference algorithm. The security can hence be guaranteed.

## 3  MOTIVATIONS AND CHALLENGES

In this section, we discuss the key observations that motivate MPCache.

**Observation 1: the attention map of a long input sequence is usually sparse, and the KV cache of historical tokens demonstrates different impacts over the downstream decoding.** We show the attention map of different heads and layers of LLaMA-2-7B in Figure 2 and leave visualizations of larger attention maps in Appendix C. From Figure 2, we can classify different tokens into three categories: 1) important to all tokens (IA in red box): the attention scores remain high for the entire column, e.g., 0th and 1st columns in Figure 2(a), indicating these tokens are important for the generation of all downstream tokens and hence, need to be always preserved; 2) un-important to all tokens (UIA in blue box): the attention scores remain low for the entire column, e.g., 2nd and 3rd columns in Figure 2(a), indicating these tokens can be discarded without impacting the downstream decoding; 3) important to certain tokens (IC in orange box): the attention scores vary for different tokens,

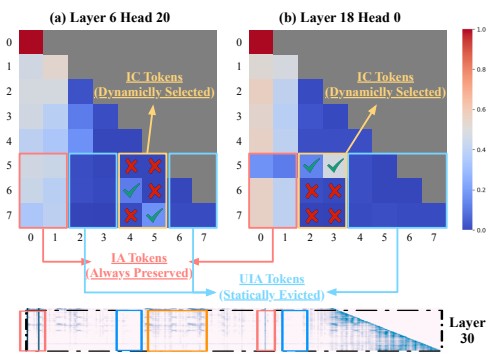

Figure 2: (Upper) token types in attention maps where ✓ means the token is selected and ✗ means the token is not selected. (Lower) three types can be observed in the attention map with more tokens.

e.g., 4th and 5th columns in Figure 2(a), indicating these tokens impact a subset of downstream tokens, and hence, cannot be directly pruned.

We verify the observation on LLaMA-2-7B with different input sequence lengths. As shown in Figure 3(a), almost 60% tokens can be statically evicted while preserving the LLM performance. While further pruning the remaining KV cache starts to degrade the LLM performance, as shown in Figure 3(b), in each decoding step, only less than 20% of the remaining tokens contribute to the decoding. *The above observation motivates us to statically evict the KV cache of UIA tokens and dynamically select a subset of IC tokens in each decoding step.*

**Observation 2: dynamic KV cache selection incurs non-negligible overhead in MPC.** While dynamic KV cache selection reduces the attention computation cost, it incurs non-negligible overhead due to MPC-unfriendly operations. In Figure 1(c), we show the extra overhead when 5% tokens are dynamically selected. The MPC-unfriendly operations mainly include:

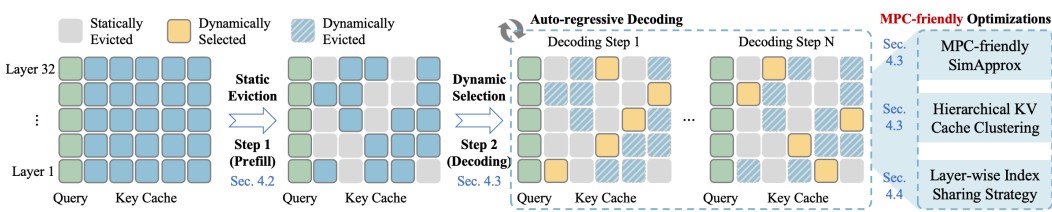

Figure 4: Overview of our proposed MPCache.

- Similarity computation (Algorithm 1 line # 1): cosine similarity is widely used for similarity measurement, which requires computing the multiplication between the current query with the key cache of all previous tokens;

- Top-$k$ selection (Algorithm 1 line # 2): to compute the indices of relevant tokens, top-$k$ is usually inevitable (Zhang et al., 2024d; Ge et al., 2023; Zhao et al., 2024; Yang et al., 2024). Unlike plaintext inference, top-$k$ selection in MPC involves frequent comparison protocol, which incurs high latency and communication cost (Rathee et al., 2020).

- Token gathering (Algorithm 1 line # 3): after the top-$k$ selection, the KV cache of selected tokens is gathered based on the indices. Unlike plaintext inference, such gathering protocol in MPC is much more inefficient since both KV cache and indices are ciphertexts. Therefore, as described in Algorithm 2, each index is first converted to a one-hot vector and then multiplied with the KV cache, requiring repetitively invoking MPC-unfriendly comparison protocols.

Inspired by token-wise locality (Liu et al., 2023; Zhu et al., 2023), *our key insight is to group the adjacent tokens into clusters*, which can reduce the complexity of dynamic selection in proportion to the cluster size. However, this introduces extra questions on how to measure the similarity between a cluster and the current query, how to build the cluster, etc, which is discussed in Section 4.3.

**Observation 3: adjacent layers share similar top-$k$ ranking of KV cache, providing an extra opportunity for efficiency optimization.** Due to the residual, we hypothesize adjacent layers may share a similar top-$k$ ranking of the KV cache. To verify the assumption, we define commonality score to measure the ratio of common top-$k$ indices of $m$ adjacent layers as below:

$$\frac{1}{k(L-m)} \sum_{l=1}^{L-m} \left| \bigcap_{i=l}^{l+m} \mathbf{idx}_i[:k] \right|, \tag{1}$$

where $\mathbf{idx}_i[:k]$ denotes the set of top-$k$ indices for $i$-th layer, $L$ is the number of layers, and $|\cdot|$ counts the number of elements in a set. As shown in Figure 3(c), adjacent layers demonstrate a high similarity of top-$k$ indices, which indicates the query tends to focus on the KV cache of the similar tokens. The similarity score reduces when $m$ is large, which motivates us to share the indices of selected tokens among adjacent layers to trade off efficiency and performance.

## 4 MPCACHE: AN MPC-FRIENDLY PRIVATE LLM INFERENCE FRAMEWORK

### 4.1 OVERVIEW OF MPCACHE

**Framework.** Driven by the observations, we propose an MPC-friendly KV cache eviction framework, dubbed MPCache. The overview is shown in Figure 4, and it consists of two steps: 1) look-once static eviction during the prefill stage to discard the UIA tokens (Section 4.2); 2) query-aware dynamic selection during the decoding stage to choose only a small subset of the remaining IC tokens for sparse attention (Section 4.3). A series of MPC-friendly optimizations are proposed to reduce the overhead of dynamic selection. The pseudocode is shown in Algorithm 3 in Appendix D.

**Symbol definition.** For clarity, we summarize the symbols used in this section. We define $L$ as the number of layers, $H$ as the number of attention heads, $T$ as the number of tokens, $d$ as the embedding dimension, $s$ as the cluster size, and $C$ as the number of clusters.

### 4.2 STEP 1: LOOK-ONCE STATIC KV CACHE EVICTION ALGORITHM

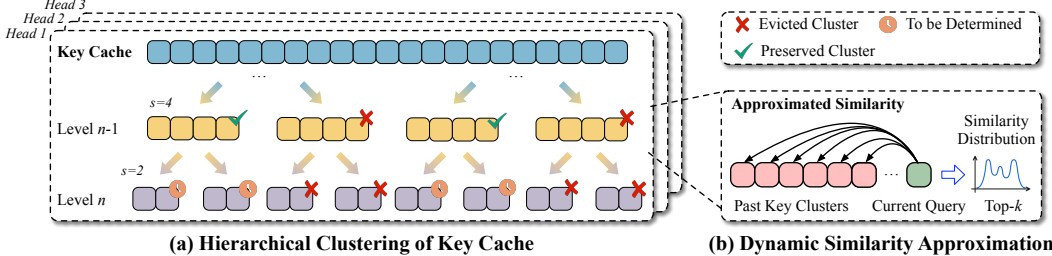

Figure 6: Hierarchical and dynamic KV cache clustering and selection procedure.

To prune the KV cache of UIA tokens as observed in Section 3, we use static eviction during the prefill stage. To measure the token importance and identify UIA tokens, we compute the attention map and then, accumulate the attention scores for each token. Similar to Zhang et al. (2024d); Liu et al. (2024c); Li et al. (2024), we find it is sufficient to only sum up the scores of the last 20% tokens in the prompt. Then, we rank the accumulated attention scores to select the top-$\gamma$ KV cache with the highest scores and discard the rest UIA tokens.

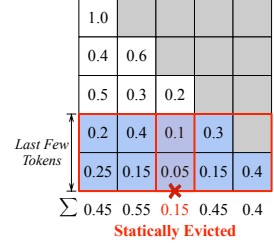

Figure 5: The illustration of static eviction.

*Protocol complexity analysis.* Compared to the baseline computation of the prefill stage, static eviction only involves accumulating the attention scores, which are local without any communication, and a top-$\gamma$ selection. Because the static eviction is performed only once, the cost of top-$\gamma$ selection can be amortized by the entire generation process, and hence, becomes negligible. Meanwhile, with UIA tokens pruned, the efficiency of the dynamic selection process can be improved for each generation step. Hence, the static eviction algorithm helps to improve the overall efficiency.

### 4.3 STEP 2: MPC-FRIENDLY DYNAMIC KV CACHE SELECTION ALGORITHM

To reduce the overhead of dynamic token selection as shown in Figure 1(c), we propose to group the KV cache of adjacent tokens into clusters as shown in Figure 6. The most important question is *"how to aggregate the information of a cluster and measure the importance of each cluster accurately and efficiently?"*

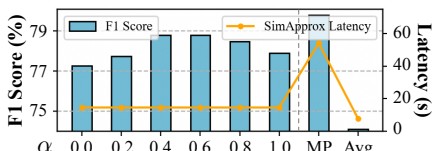

Figure 7: Comparison among maximum dot product (MP), average, and our method with different $\alpha$'s on TriviaQA.

**MPC-friendly similarity approximation with clustering.** A naive method for similarity approximation is to compute the average of the key cache within a cluster and directly compute the cosine similarity with the average. However, as shown in Figure 7, the naive approach incurs large performance degradation. *Our intuition is the approximation should preserve the impact of important tokens as much as possible.* Hence, we use the maximum dot product between the query and the key cache cluster. Specifically, given a query $\mathbf{q} \in \mathbb{R}^{1 \times d}$, a key cache cluster of $s$ tokens $\mathbf{K}_c \in \mathbb{R}^{s \times d}$, the similarity can be designed as

$$\text{SimApprox}(\mathbf{q}, \mathbf{K}_c) = \max_{\mathbf{k} \in \mathbf{K}_c} \mathbf{q} \cdot \mathbf{k} = \max_{\mathbf{k} \in \mathbf{K}_c} \sum_{i=0}^{d-1} \mathbf{q}_i \mathbf{k}_i \leq \sum_{i=0}^{d-1} \max_{\mathbf{k} \in \mathbf{K}_c} \mathbf{q}_i \mathbf{k}_i, \quad (2)$$

where we obtain the upper bound of similarity. We further have

$$\max_{\mathbf{k} \in \mathbf{K}_c} \mathbf{q}_i \mathbf{k}_i = \begin{cases} \mathbf{q}_i \max_{\mathbf{k} \in \mathbf{K}_c} \mathbf{k}_i & \text{if } \mathbf{q}_i \geq 0, \\ \mathbf{q}_i \min_{\mathbf{k} \in \mathbf{K}_c} \mathbf{k}_i & \text{if } \mathbf{q}_i < 0. \end{cases} \quad (3)$$

Define $\mathbf{r}^{\max}$ and $\mathbf{r}^{\min}$, where $\mathbf{r}_i^{\max} = \max_{\mathbf{k} \in \mathbf{K}_c} \mathbf{k}_i$ and $\mathbf{r}_i^{\min} = \min_{\mathbf{k} \in \mathbf{K}_c} \mathbf{k}_i$. Then, we have

$$\text{SimApprox}(\mathbf{q}, \mathbf{K}_c) \leq \sum_{i=0}^{d-1} \max_{\mathbf{k} \in \mathbf{K}_c} \mathbf{q}_i \mathbf{k}_i = \sum_{i=0}^{d-1} \max(\mathbf{q}_i \mathbf{r}_i^{\max}, \mathbf{q}_i \mathbf{r}_i^{\min}). \quad (4)$$

Table 2: The complexity analysis of token gathering protocol where $k_1 = 0.25T, k_2 = 0.25C$.

| | Bit Width | # Comparison | Lat. | Comm. | Example Lat. | Example Comm. |
|---|---|---|---|---|---|---|
| Baseline Protocol | $\log T$ | $T$ | $\mathcal{O}(T \log T)$ | $\mathcal{O}(k_1 T \log T)$ | 4.780s | 416.0MB |
| MPCache (ours) | $\log C$ | $C$ | $\mathcal{O}(C \log C)$ | $\mathcal{O}(k_2 C \log C)$ | 0.065s | 1.125MB |
| Improvement | $\frac{\log T}{\log C} \times$ | $\frac{T}{C} \times$ | $\frac{T \log T}{C \log C} \times$ | $\frac{k_1 T \log T}{k_2 C \log C} \times$ | 73.5× | 369.8× |

*Protocol complexity analysis.* During the decoding stage, $\mathbf{r}^{\max}$ and $\mathbf{r}^{\min}$ of each cluster only need to be computed once. Hence, the computation cost can be amortized and become negligible. However, for each generation step, we still need to compute $\mathcal{O}(LCd)$ multiplications, i.e., $\mathbf{q}_i \mathbf{r}_i^{\max}$ and $\mathbf{q}_i \mathbf{r}_i^{\min}$, as well as $\mathcal{O}(LCd)$ max operations in Equation (4), which still incur non-negligible overhead.

**Linearization and Reordering.** To avoid the MPC-unfriendly max operation in Equation (4), we further propose to approximate the similarity score as below:

$$\text{SimApprox}(\mathbf{q}, \mathbf{K}_c) \approx \sum_{i=0}^{d-1} \alpha \cdot \mathbf{q}_i \mathbf{r}_i^{\max} + (1 - \alpha) \cdot \mathbf{q}_i \mathbf{r}_i^{\min}, \tag{5}$$

where $\alpha \in [0, 1]$ is a hyperparameter. As can be observed, when $\alpha = 1$, $\mathbf{q}_i \mathbf{r}_i^{\max}$ is always selected while $\mathbf{q}_i \mathbf{r}_i^{\min}$ is always selected when $\alpha = 0$. After the linearization, there is an opportunity to further reduce the multiplications by reordering the computation as

$$\sum_{i=0}^{d-1} \alpha \cdot \mathbf{q}_i \mathbf{r}_i^{\max} + (1 - \alpha) \cdot \mathbf{q}_i \mathbf{r}_i^{\min} = \sum_{i=0}^{d-1} \mathbf{q}_i \cdot (\alpha \mathbf{r}_i^{\max} + (1 - \alpha) \mathbf{r}_i^{\min}). \tag{6}$$

$\alpha \mathbf{r}_i^{\max}$ and $(1 - \alpha) \mathbf{r}_i^{\min}$ are first added up without introducing extra communication, and the multiplication with $\mathbf{q}_i$ is reduced by 2×. Compared with the maximum dot product in Figure 7, our method significantly reduces the cost while maintaining the performance. We empirically choose $\alpha = 0.6$, and leave more discussions to Appendix F and a theoretical analysis to Appendix G.

*Protocol complexity analysis.* MPCache reduces the number of max operations from $\mathcal{O}(LCd)$ to 0 and reduce the multiplication complexity by 2×. Clustering also benefits the token gathering protocol: 1) the number of comparisons in one-hot vector conversion is reduced by $\frac{T}{C} \times$; 2) the bit width of one-hot vector is reduced by $\frac{\log T}{\log C} \times$. Table 2 shows an example of selecting top-25% tokens with $T = 1024, C = 64$, and can be observed that the overhead is drastically reduced.

**Hierarchical KV cache clustering.** Another question is *"how to build the KV cache cluster?"* Since larger cluster sizes have higher selection efficiency at the cost of worse performance, our key insight is to trade off the selection overhead and model performance. Inspired by hierarchical reinforcement learning (Xu et al., 2023), we propose to cluster the KV cache of adjacent tokens with a hierarchical structure as shown in Figure 6 that conducts coarse-grained (with larger cluster size) to fine-grained (with smaller cluster size) selection. Generally, we divide the KV cache into $n$ levels and progressively select the clusters level by level from the coarse-grained one. Then, at the fine-grained level, we only need to select from the remaining clusters, thereby reducing the selection complexity. Hierarchical structure, including the cluster size and selection ratios at different levels, can influence the performance-efficiency trade-off, which is discussed in Section 5.4.

## 4.4 LAYER-WISE INDEX SHARING FOR FURTHER EFFICIENCY OPTIMIZATION

To leverage the observation that adjacent layers share similar top-$k$ ranking of KV cache, we propose a layer-wise index sharing strategy that enables adjacent layers to share the same selected token indices to further reduce the cost of dynamic selection. Since two adjacent layers show the highest commonality score in Figure 3(c), we choose to share the indices between two adjacent layers. In Figure 8, we observe the first two layers have a low commonality score while other layers have higher scores due to the residual, so we do not apply sharing to the first two layers. Layer-wise index sharing effectively reduces the extra overhead introduced by dynamic selection. We discuss how the number of adjacent layers affects the trade-off in Section 5.4.

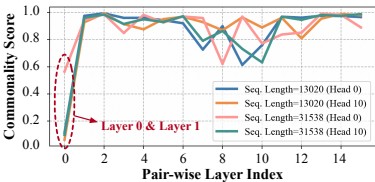

Figure 8: Commonality score between two adjacent layers on LLaMA-2-7B.

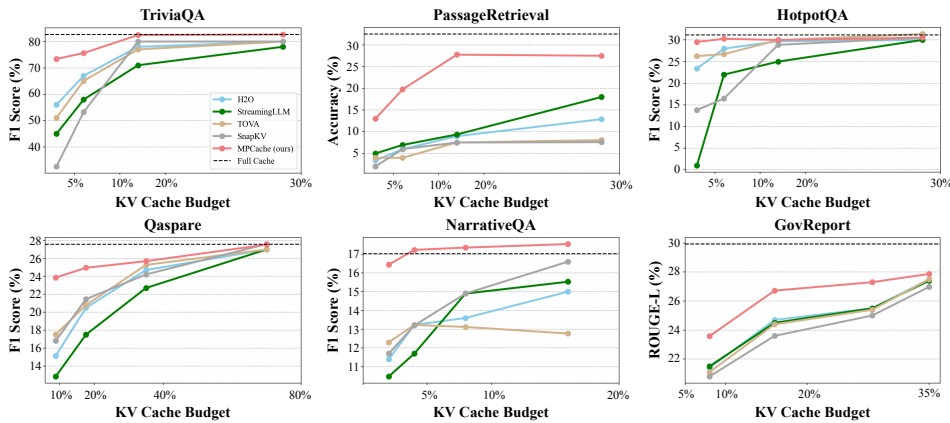

Figure 9: Comparison with fixed-pattern and static KV cache eviction baselines.

# 5 EMPIRICAL EVALUATION

## 5.1 EXPERIMENTAL SETUPS

**Models and datasets.** Our experiments are based on LongChat-7B-V1.5-32K (Li et al., 2023) on LongBench (Bai et al., 2023)[1]: HotpotQA (Yang et al., 2018), NarrativeQA (Kočiskỳ et al., 2018), Qasper (Dasigi et al., 2021), GovReport (Huang et al., 2021), TriviaQA (Joshi et al., 2017), and PassageRetrieval (Bai et al., 2023). We also apply our method to LLaMA-2-7B/13B (Touvron et al., 2023) on 5-shot XSUM (Narayan et al., 2018) and LLaMA-3-8B-Instruct (Dubey et al., 2024) on LongBench. To save GPU memory when processing long-context tasks, we leverage FlashAttention (Dao et al., 2022) during the prefill stage.

**Baselines.** For comparison, we choose prior-art static and dynamic KV cache eviction baselines, including H2O (Zhang et al., 2024d), StreamingLLM (Xiao et al., 2023), TOVA (Oren et al., 2024), SnapKV (Li et al., 2024), InfLLM (Xiao et al., 2024), and LongCache (Liu et al., 2024b). Detailed descriptions of the baselines and our setups can be found in Appendix F.

**Experimental environment.** For performance evaluation, our experiments are conducted based on LongBench on an NVIDIA A100 80GB GPU. For efficiency evaluation, our experiments are based on Secretflow (SPU (Ma et al., 2023) V0.9.1) and follow the protocols of PUMA (Dong et al., 2023)[2]. We optimize the top-$k$ protocol in Secretflow with computation parallelization. The latency is evaluated under the LAN setup (Rathee et al., 2020). We evaluate the efficiency using GPT-2 and LLaMA-2, and since securely evaluating a full-size 7B model in SPU exceeds our hardware resources, we set a smaller hidden dimension of 1024 in our evaluation.

## 5.2 PERFORMANCE EVALUATION

In Figure 9 and Table 4, we comprehensively compare MPCache with prior-art KV cache eviction methods and make the following observations: **1) comparison with fixed-pattern and static algorithms.** MPCache consistently outperforms prior-art methods, including H2O, StreamingLLM, TOVA, and SnapKV across different datasets. These methods statically discard the tokens while MPCache dynamically selects a subset of tokens based on the current queries. MPCache shows decent scalability to different KV cache budgets. For example, on HotPotQA and NarrativeQA, MPCache achieves comparable performance as full cache, even only ~5% KV cache preserved; **2) comparison with dynamic algorithms.** MPCache achieves comparable and even better performance compared with InfLLM and LongCache. For example, on NarrativeQA, MPCache achieves 1.32× and 2.39× latency reduction with a higher F1 score compared with InfLLM and LongCache, respectively; **3) scalability of MPCache.** We extend our method to LLaMA-2-13B in Figure 3 and LLaMA-3-8B-Instruct in Table 5, demonstrating the superior performance of MPCache.

[1] https://github.com/THUDM/LongBench
[2] https://github.com/secretflow/spu

Table 4: Comparison with dynamic eviction baselines on different datasets and budgets. "(a×)" means MPCache achieves a× efficiency improvement compared with baselines.

| Dataset | Cache Budget | InfLLM | | LongCache | | MPCache (ours) | |
|---|---|---|---|---|---|---|---|
| | | Perf. (%)↑ | Lat. (s)↓ | Perf. (%)↑ | Lat. (s)↓ | Perf. (%)↑ | Lat. (s)↓ |
| HotpotQA | Full | 31.16 | 75.52 | 31.16 | 75.52 | 31.16 | 75.52 |
| | 5% | 28.20 | 51.64 (1.30×) | 24.31 | 89.46 (2.24×) | 30.27 | 39.85 |
| | 10% | 29.01 | 68.04 (1.28×) | 24.69 | 123.1 (2.30×) | 30.05 | 53.32 |
| TriviaQA | Full | 82.67 | 75.52 | 82.67 | 75.52 | 82.67 | 75.52 |
| | 5% | 75.65 | 51.64 (1.38×) | 59.85 | 89.46 (2.39×) | 75.61 | 37.37 |
| | 10% | 82.75 | 68.04 (1.34×) | 60.56 | 123.1 (2.43×) | 82.45 | 50.75 |
| NarrativeQA | Full | 17.02 | 75.52 | 17.02 | 75.52 | 17.02 | 75.52 |
| | 5% | 12.80 | 47.74 (1.32×) | 14.65 | 86.42 (2.39×) | 17.23 | 36.13 |
| | 10% | 13.74 | 63.49 (1.28×) | 15.69 | 121.4 (2.45×) | 17.35 | 49.46 |
| PassageRetrieval | Full | 32.50 | 75.52 | 32.50 | 75.52 | 32.50 | 75.52 |
| | 5% | 6.161 | 51.64 (1.15×) | 21.42 | 89.46 (1.99×) | 19.75 | 44.82 |
| | 10% | 8.872 | 68.04 (1.16×) | 24.92 | 123.1 (2.10×) | 27.75 | 58.47 |
| Qasper | Full | 27.58 | 75.52 | 27.58 | 75.52 | 27.58 | 75.52 |
| | 8% | 20.53 | 64.52 (1.45×) | 24.53 | 136.9 (3.08×) | 23.86 | 44.39 |
| | 16% | 23.90 | 72.84 (1.33×) | 26.07 | 225.9 (4.12×) | 24.95 | 54.77 |

Table 5: Extension to LLaMA-3-8B-Instruct on with an average KV cache size of 2048.

| Method | Qasper (F1 Score) | MultiFieldQA (F1 Score) | HotpotQA (F1 Score) | 2WikiMultihopQA (F1 Score) | MuSique (F1 Score) | TriviaQA (F1 Score) | TREC (Accuracy) | SAMSum (Rouge-L) |
|---|---|---|---|---|---|---|---|---|
| Full Cache | 29.75 | 41.12 | 45.55 | 35.87 | 22.35 | 90.56 | 73.0 | 41.88 |
| SnapKV | 25.78 | 38.13 | 40.12 | 32.01 | 16.86 | 83.22 | 70.0 | 31.75 |
| H2O | 26.85 | 39.54 | 44.30 | 32.92 | 21.09 | 90.56 | 53.0 | 41.84 |
| MPCache (ours) | 29.45 | 40.30 | 44.32 | 35.91 | 22.66 | 90.43 | 73.0 | 42.42 |

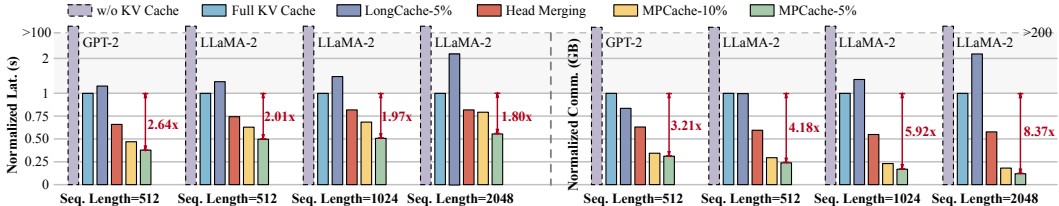

Figure 10: Evaluation on per-token generation latency and communication.

## 5.3 INFERENCE EFFICIENCY EVALUATION

In Figure 10, we benchmark the generation efficiency with different sequence lengths ranging from 512 to 2048. We compare MPCache with model without KV cache, with full KV cache, LongCache, and head merging (Rathee et al., 2024; Bian et al., 2021). From the results, we make the following observations: 1) KV cache is crucial for private LLM inference since it avoids re-computation of the KV cache of the previous tokens. As shown in the purple bar, the overhead increases by hundreds of times compared with using

Table 3: Comparison of LLMs with different parameter scales on XSUM.

| Budget | 10% | | 5% | |
|---|---|---|---|---|
| Scale | 7B↑ | 13B↑ | 7B↑ | 13B↑ |
| Full Cache | 11.90 | 13.60 | 11.90 | 13.60 |
| H2O | 10.50 | 13.24 | 4.886 | 9.081 |
| MPCache (ours) | 11.10 | 13.44 | 10.08 | 13.08 |

the KV cache; 2) compared with full KV cache on LLaMA-2, MPCache achieves $1.59 \sim 2.01\times$, $1.46 \sim 1.97\times$, and $1.26 \sim 1.8\times$ latency reduction and $3.39 \sim 4.18\times$, $4.33 \sim 5.92\times$, and $5.51 \sim 8.37\times$ communication reduction with different sequence lengths, respectively; 3) compared with LongCache which dynamically selects tokens without static eviction and clustering on LLaMA-2, MPCache even achieves $3.85\times$ and $19.47\times$ latency and communication reduction, respectively. We further discuss the 2PC protocol Lu et al. (2023) in Section 5.4.

## 5.4 ABLATION STUDY OF MPCACHE

**Effectiveness of different optimizations.** In Figure 11, we demonstrate the effectiveness of our proposed optimizations by adding them step by step on LLaMA-2-7B with a sequence length of 1024 and static eviction ratio of 75%. We make the following observations: 1) directly applying

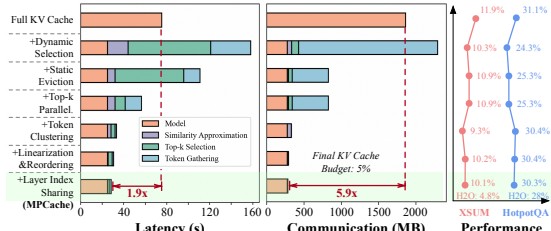

Figure 11: Step-by-step ablation study of MPCache.

Table 6: Different hierarchical structures with a dynamic selection ratio of 20%.

| Level 1 Coarse-grained | Level 2 Fine-grained | F1 Score (%) | Comm. (MB) |
|---|---|---|---|
| s32(0.9) | s16(0.22) | 29.6 | 163.5 |
| s32(0.7) | s16(0.28) | 30.1 | 144.0 |
| s32(0.5) | s16(0.40) | 30.2 | 140.2 |
| s32(0.3) | s16(0.67) | 29.2 | 108.8 |
| s64(0.9) | s16(0.22) | 29.5 | 158.1 |
| s64(0.7) | s16(0.28) | 29.3 | 110.1 |
| s64(0.5) | s16(0.40) | 29.1 | 104.9 |
| s64(0.3) | s16(0.67) | 29.0 | 69.12 |

dynamic selection, e.g., LongCache to private LLM inference does not provide the expected efficiency improvement and even increases both latency and communication; 2) after static eviction, latency and communication of dynamic selection are reduced by $1.42\times$ and $2.76\times$, respectively. 3) our MPC-friendly optimizations, including clustering, linearization, reordering, and layer index sharing further reduce the extra overhead introduced by dynamic selection without sacrificing the model performance; 4) MPCache eventually achieves $1.9\times$ and $5.9\times$ latency and communication reduction, respectively, and achieves better performance compared with H2O.

**Effect of hierarchical structure.** To trade off the model performance and dynamic selection overhead, we evaluate different hierarchical structures on HotpotQA. Specifically, we choose different cluster sizes $s$ and selection ratios at different levels (e.g., $s32(0.7)$ means selecting 70% clusters with $s = 32$). From Figure 6, we make the following conclusions: 1) when the gap between two levels increases or the coarse-grained selection ratio decreases, the overhead becomes lower and the performance exhibits a downward trend; 2) appropriate course-grained selection may help improve the performance, e.g., the ratio changes from 90% to 50% with $s = 32$;

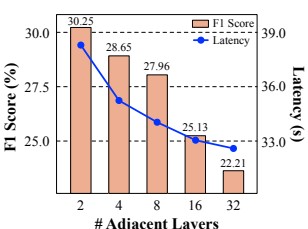

Figure 12: Effect of # adjacent layers.

**Effect of the number of adjacent layers for layer index sharing.** In response to Section 3, we evaluate the trade-off between the number of adjacent layers for layer-wise index sharing and model performance on HotpotQA in Figure 12. As observed, when the number of adjacent layers increases, the latency is reduced at the cost of the performance degradation.

**Discussion on 2PC protocol.** We evaluate the 2PC efficiency in Figure 13. It is observed that MPCache achieves $1.63\times$ and $1.79\times$ latency and communication reduction compared with the full cache, and $2.58\times$ and $2.48\times$ latency and communication reduction compared with Long-Cache. Since the multiplication communication in 2PC is larger than in 3PC, the cost of similarity approximation becomes higher. We can use random projection (Johnson et al., 1986) to reduce the multiplication dimensionality, and we leave the research as our future work.

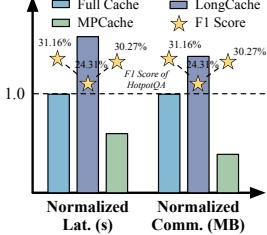

Figure 13: Extension to 2PC protocol.

**Additional results.** We present more experimental results, including the effect of $\alpha$, the necessity of KV cache, and the comparison with average-based similarity approximation in Appendix F.

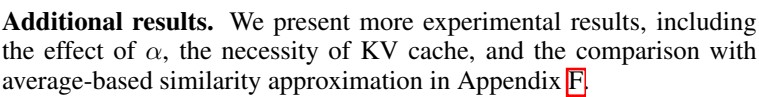

## 6 CONCLUSION

In this work, we propose an MPC-friendly KV cache eviction framework dubbed MPCache, that enables accurate and efficient private LLM inference. MPCache is a two-step framework combining static eviction and dynamic selection. To reduce the heavy overhead of dynamic selection, we propose a series of MPC-friendly optimizations. Extensive evaluations demonstrate that MP-Cache consistently outperforms prior-art KV cache eviction baselines across different generation tasks and significantly reduces both latency and communication.

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
