## A DETAILED BACKGROUND AND RELATED WORKS

### A.1 PRIVATE LLM INFERENCE

Recently, private LLM inference has attracted an increasing amount of research attention. PUMA (Dong et al., 2023) proposes a series of 3PC protocols for both linear and non-linear functions to support private LLM inference, even under the scale of LLaMA-7B. BumbleBee (Lu et al., 2023) proposes homomorphic encryption (HE)-based protocols that enable the multiplication of large matrices and efficient protocols for non-linear functions similar to PUMA. CipherGPT (Hou et al., 2023) uses subfield vector oblivious linear evaluation (sVOLE) to reduce the communication of MatMuls significantly. BOLT (Pang et al., 2023) proposes a baby-step giant-step (BSGS) strategy that reduces the number of rotations on ciphertexts. SIGMA (Gupta et al., 2023) achieves private GPT inference with function secret sharing (FSS) and accelerates the computation on GPUs. PermLLM (Zheng et al., 2024) proposes an efficient protocol for non-linear functions based on the random permutation. However, they still incur significant overhead, especially on long sequences.

There are also works directly replacing expensive non-linear functions, e.g., Softmax and GeLU with MPC-friendly operations. For instance, MPCFormer (Li et al., 2022) simplifies Softmax by replacing exponential with MPC-friendly quadratic operation. MPCViT (Zeng et al., 2023) proposes to selectively replace exponential in Softmax with ReLU through neural architecture search. (Liu & Liu, 2023) directly uniformly replaces GeLU with ReLU and replaces exponential in Softmax with ReLU. Although these methods achieve high inference efficiency, they cannot avoid finetuning or retraining to preserve the model performance, making LLM development impractical. In conclusion, the above works still suffer from heavy overhead. Moreover, these works always handle full-length contexts during the LLM generation process. To solve this problem, our work aims to compress the KV cache with MPC-friendly optimizations. Note that our proposed method can also be applied to different protocol frameworks for efficiency improvement.

### A.2 KV CACHE COMPRESSION

When tackling the LLM generation tasks, especially in long-context scenarios, the KV cache in the attention module becomes the most significant bottleneck due to the increasing sequence length. Therefore, how to effectively reduce the size of the KV cache is a high priority. System-level optimizations such as FlashAttention (Dao et al., 2022), FlashAttention-2 (Dao, 2023), FlashAttention-3 (Shah et al., 2024), and PagedAttention (Kwon et al., 2023) have been proposed to alleviate the problem. Meanwhile, many recent research efforts have been devoted to algorithm-level optimizations. For example, *quantization* methods (Hooper et al., 2024; Zhang et al., 2024b; Kang et al., 2024; He et al., 2024; Liu et al., 2024d) have been proposed to compress KV cache to $1 \sim 4$ bits, *linear attention mechanisms* (Kitaev et al., 2020; Zeng et al., 2023; Choromanski et al., 2020) have been proposed to reduce the quadratic complexity w.r.t. the sequence length. In this work, we follow FlashAttention (Dao et al., 2022) to save the GPU memory and focus on another research line of algorithm-level optimization called *KV cache eviction*, which is designed to reduce the number of tokens and enable sparse attention without extra finetuning.

KV cache eviction can be roughly categorized into 3 classes: 1) *fixed-pattern algorithm:* the position of important tokens is pre-defined before inference and remains consistent across decoding steps. However, this algorithm is not flexible for different LLMs and contexts (Xiao et al., 2023; Beltagy et al., 2020); 2) *static algorithm:* tokens are statically discarded and cannot be recovered in the subsequent decoding steps. This algorithm is usually efficient but suffers from significant performance degradation when the compression ratio is high (Zhang et al., 2024d; Liu et al., 2024c; Ge et al., 2023); 3) *dynamic algorithm:* tokens are dynamically selected across different decoding steps. This algorithm is much more flexible but the dynamic selection usually involves more expensive operations (Xiao et al., 2024; Liu et al., 2024b; Tang et al., 2024b).

Here, we introduce recent works of KV cache eviction. StreamingLLM (Xiao et al., 2023) proposes to keep a few initial tokens along with the recent tokens to recover the long-context performance. RazorAttention (Tang et al., 2024a) theoretically analyzes the scope of effective attention vision for each head. Scissorhands (Liu et al., 2024c), H2O (Zhang et al., 2024d), ALISA (Zhao et al., 2024), spAtten (Wang et al., 2021), and TOVA (Oren et al., 2024) use the accumulated attention score of the historical tokens to preserve a small subset of KV cache. FastGen (Ge et al., 2023) proposes

to allocate different eviction policies for different heads based on the profiling result of the prompt. SnapKV (Li et al., 2024) and LOOK-M (Wan et al., 2024) select important tokens for each attention head based on the attention weights of prompts. PyramidKV (Zhang et al., 2024c), PyramidInfer (Yang et al., 2024), and SqueezeAttention (Wang & Gan, 2024) consider allocating different KV cache budgets for different layers. InfLLM (Xiao et al., 2024) and LongCache (Liu et al., 2024b) propose to dynamically select tokens based on the relationship between the current query and the key cache of previous tokens. RetrievalAttention (Liu et al., 2024a) establishes connections from the query to its nearest keys and the decoding query can first search its nearest query and then obtain the most relevant key vectors. LazyLLM (Fu et al., 2024) introduces an aux cache to enable selective KV cache eviction. Keyformer (Adnan et al., 2024) finds that the distribution after token pruning becomes uneven and proposes to smooth the distribution. The above works explicitly rely on the attention weights such that they are incompatible with FlashAttention (Luohe et al., 2024). To get rid of the dependency of attention weights, SirLLM (Yao et al., 2024) uses token entropy while Devoto et al. (2024) uses the L2-norm of the key cache to measure the token importance. However, these works are not designed or optimized for MPC since they either statically evict tokens that cause significant performance degradation, or dynamically select tokens, introducing more complex and MPC-unfriendly operations. We quantitatively compare existing methods in Table 1.

### A.3 GENERATIVE LLM INFERENCE IN AUTOREGRESSIVE-STYLE

The generative inference procedure of LLM is generally in autoregressive-style such as GPT-2 (Radford et al., 2019) and LLaMA (Touvron et al., 2023), and mainly consists of two stages: 1) the prefill (prompt) stage and 2) the decoding (generation) stage.

The prefill stage serves as the first step of generation. LLM takes a prompt sequence as input and generates a key-value cache (KV cache) for each layer as

$$\mathbf{O}_{\text{prompt}} = \text{Softmax}(\mathbf{Q}_{\text{prompt}} \cdot \mathbf{K}_{\text{prompt}}^{\top}/\sqrt{d}) \cdot \mathbf{V}_{\text{prompt}}, \tag{7}$$

where $\mathbf{Q}_{\text{prompt}} \in \mathbb{R}^{H \times T \times d}$ denotes the input query and $\mathbf{K}_{\text{prompt}} \in \mathbb{R}^{H \times T \times d}, \mathbf{V}_{\text{prompt}} \in \mathbb{R}^{H \times T \times d}$ denote the key and value tensor, respectively. After the prefill stage, the KV cache is generated as $\mathbf{K}_{\text{cache}} \leftarrow \mathbf{K}_{\text{prompt}}$ and $\mathbf{V}_{\text{cache}} \leftarrow \mathbf{V}_{\text{prompt}}$. KV cache retains previously computed key-value pairs, eliminating the need for costly re-computation of previous key and value vectors (Ott, 2019). Note that each layer is equipped with its unique KV cache and the generated KV cache is the foundation for the dowmstreaming decoding stage.

The decoding stage uses and updates the stored KV cache to generate new tokens step-by-step. First, the KV caches are updated by concatenating new $\mathbf{k} \in \mathbb{R}^{H \times 1 \times d}$ and $\mathbf{v} \in \mathbb{R}^{H \times 1 \times d}$ as

$$\mathbf{K}_{\text{cache}} \leftarrow [\mathbf{K}_{\text{cache}}||\mathbf{k}], \ \mathbf{V}_{\text{cache}} \leftarrow [\mathbf{V}_{\text{cache}}||\mathbf{v}], \tag{8}$$

where $[\cdot||\cdot]$ denotes tensor concatenation. Therefore, the attention can be computed as

$$\mathbf{O}_{\text{dec}} = \text{Softmax}(\mathbf{Q}_{\text{dec}} \cdot \mathbf{K}_{\text{cache}}^{\top}/\sqrt{d}) \cdot \mathbf{V}_{\text{cache}}, \tag{9}$$

where $\mathbf{O}_{\text{dec}}$ denotes the current query. The attention output $\mathbf{O}_{\text{dec}} \in \mathbb{R}^{1 \times d}$ is then sent to the multi-layer perceptron (MLP) layer for the subsequent computation.

## B MPC PROTOCOL DESCRIPTIONS

### B.1 THREAT MODEL AND SECURITY

Consistent with previous works (Mohassel & Rindal, 2018; Li et al., 2022; Dong et al., 2023), MP-Cache adopts an *honest-but-curious* (a.k.a., honest-but-curious) security model in honest-majority (Lindell & Pinkas, 2009) where parties follow the protocol specifications but may also try to learn more from the information than allowed. In our threat model, we assume all the parties are aware of the LLM architecture and number of pruned tokens, which is consistent with HEPrune (Zhang et al., Seesaw (Li et al., SENet (Kundu et al. (2023), SNL (Cho et al. (2022), etc. We argue that this information does not compromise the client's data or inference results, nor does it enable the client to access the model's parameters.

## B.2 2PC PROTOCOL

We follow the 2PC protocols proposed in BumbleBee (Lu et al., 2023). The protocols are built based on the 2-out-of-2 additive secret sharing (SS), where secret value $x \in \mathbb{Z}_{2^\ell}$ is shared by two random values $x_0, x_1 \in \mathbb{Z}_{2^\ell}$ such that $x = x_0 + x_1 \pmod{2^\ell}$, and party $P_i$ gets $x_i$ (denoted as $[\![x]\!]$). SS supports both addition and multiplication on the secret shares. Without special declaration, we compute in $\mathbb{Z}_{2^\ell}$ and omit $\pmod{2^\ell}$ for brevity. In the case of $\ell > 1$ (e.g., $\ell = 64$) which support arithmetic operations (e.g., $+$, $-$, and $\cdot$), we refer to this type as *arithmetic sharing*. *Boolean sharing* refers to $\ell = 1$ where $(+, -)$ and $\cdot$ are replaced by bit-wise $\oplus$ and $\wedge$, respectively.

- *Addition.* $[\![x + y]\!]$ can be computed as $(x_0 + y_0, x_1 + y_1)$, where $P_i$ can compute its share locally.

- *Multiplication.* We write the multiplication of two shared values as $[\![xy]\!] = (x_0 + x_1)(y_0 + y_1) = x_0 y_0 + x_1 y_1 + x_0 y_1 + x_1 y_0$ where two cross terms $x_0 y_1, x_1 y_0$ can be computed using hormomorphic encryption (HE).

Lu et al. (2023) uses HE scheme that is based on ring learning-with-error (RLWE). For more details about the 2PC protocol, please refer to Lu et al. (2023); Ma et al. (2023).

## B.3 3PC PROTOCOL

We follow the 3PC protocols proposed in PUMA (Dong et al., 2023). The protocols are built based on the 2-out-of-3 replicated secret sharing (RSS), where a secret value $x \in \mathbb{Z}_{2^\ell}$ is shared by three random values $x_0, x_1, x_2 \in \mathbb{Z}_{2^\ell}$ such that $x = x_0 + x_1 + x_2 \pmod{2^\ell}$, and party $P_i$ gets $(x_i, x_{i+1})$ (denoted as $[\![x]\!]$).

Let $(c_1, c_2, c_3)$ be public constants, and $([\![x]\!], [\![y]\!])$ be two secret-shared values. The secure addition and multiplication procedures are as follows:

- *Addition.* $[\![c_1 x + c_2 y + c_3]\!]$ can be computed as $(c_1 x_0 + c_2 y_0 + c_3, c_1 x_1 + c_2 y_1, c_1 x_2 + c_2 y_2)$, where $P_i$ can compute its share locally. When $(c_1 = 1, c_2 = 1, c_3 = 0)$, we get $[\![x + y]\!]$.

- *Multiplication.* Parties follow steps: i) first, $P_i$ computes $z_i = x_i y_i + x_{i+1} y_i + x_i y_{i+1}$ locally; ii) parties then perform *re-sharing* by letting $P_i$ sends $z_i' = \alpha_i + z_i$ to $P_{i-1}$, where $\alpha_0 + \alpha_1 + \alpha_2 = 0$ ($P_i$ can generate $\alpha_i$ using pseudorandom generators with negligible overhead as Mohassel & Rindal (2018)); iii) finally, $\{(z_0', z_1'), (z_1', z_2'), (z_2', z_0')\}$ form the 2-out-of-3 replicated secret shares of $[\![xy]\!]$.

For more details about the 3PC protocol, please refer to Mohassel & Rindal (2018); Ma et al. (2023).

## B.4 TOKEN GATHERING

Token gathering is used to retrieve tokens in the KV cache based on the indices, which has the same functionality as $\text{torch.gather}(\text{tensor}, \text{indices})$ in PyTorch programming. We illustrate the overall procedure in Figure 14.

For brevity, in Algorithm 2, we show the pipeline that retrieves one token from the key cache (we also omit the head dimension for simplification). The first step is converting a ciphertext index $[\![\mathbf{id}]\!]$ into a ciphertext one-hot vector $[\![\mathbf{o}]\!] \in \mathbb{R}^{1 \times T}$ based on equal protocol $\Pi_{\text{Equal}}$, where $T$ denotes the number of tokens. Given a ciphertext key cache $[\![\mathbf{K}]\!] \in \mathbb{R}^{T \times D}$, where $D$ denotes the dimension, multiplying $[\![\mathbf{o}]\!]$ with $[\![\mathbf{K}]\!]$ (matrix-vector multiplication $\Pi_{\text{MatVec}}$) can generate an output with dimension $1 \times D$, which is the retrieved token. To extend the case to retrieve $m$ tokens, we concatenate $m$ one-hot vectors to form a matrix $[\![\mathbf{O}]\!] \in \mathbb{R}^{m \times T}$, and then multiply $[\![\mathbf{O}]\!]$ with $[\![\mathbf{K}]\!]$ (matrix-matrix multiplication $\Pi_{\text{MatMul}}$) to generate an output with dimension $m \times D$, which is the retrieved tokens.

Note that token gathering protocol is also used on the value cache, and its indices are consistent with that of the key cache.

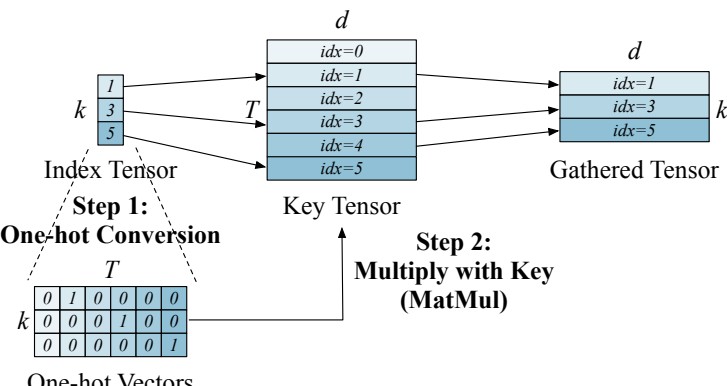

Figure 14: Illustration of token gathering procedure.

---

**Algorithm 2:** Token gathering protocol $\Pi_{\text{Gather}}$ for retrieving one token

**Input** : A ciphertext key cache $[\![\mathbf{K}]\!] \in \mathbb{R}^{T \times D}$ and a ciphertext index $[\![\mathbf{id}]\!]$.
**Output:** Key cache $[\![\mathbf{K}]\!]' \in \mathbb{R}^{1 \times D}$ with the selected token.

1 **for** $i \in [0, \dots, T-1]$ **do**
2     Parties jointly generate the one-hot vector as $[\![\mathbf{o}[i]]\!] = \Pi_{\text{Equal}}([\![\mathbf{id}]\!], i)$;
3 Parties jointly compute the retrieved key cache as $[\![\mathbf{K}]\!]' = \Pi_{\text{MatVec}}([\![\mathbf{o}]\!], [\![\mathbf{K}]\!])$;
4 **return** $[\![\mathbf{K}]\!]'$.

---

## C   OBSERVATION FROM PATTERN DISCOVERY OF LARGE ATTENTION MAPS

It is sufficient to use a few tokens within the observation window to distinguish the attention patterns since the structure of attention maps is stable in different generation steps (Liu et al., 2024c; Yang et al., 2024; Ge et al., 2023; Li et al., 2024). In Figure 15, we visualize the large attention map with hundreds of tokens on the PiQA (Bisk et al., 2020) dataset to further verify our observation in Section 3. As can be observed, there are three types of tokens defined in Section 3: 1) IA tokens in red blocks which usually appear as attention sinks mentioned in StreamingLLM (Xiao et al., 2023). 2) IC tokens in orange blocks; 3) UIA tokens in blue blocks. The pattern of attention maps motivates us to statically discard the UIA tokens which may have negligible impact on further generation, and dynamically select important tokens from IC tokens at each decoding step for sparse attention computation.

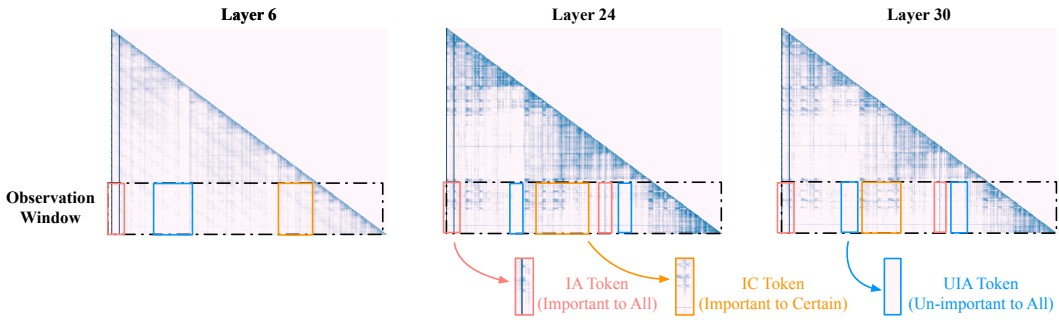

Figure 15: Attention patterns across different layers on LLaMA-7B.

## D  PSEUDOCODE OF MPCACHE ALGORITHM FRAMEWORK

We describe the algorithm flow of our MPCache in detail as shown in Algorithm 3.

---

**Algorithm 3:** KV cache eviction framework combining static and dynamic algorithm

---

**Input** : Input sequence prompt; LLM model $\mathcal{M}$; number of layers and attention heads $L$ and $H$; dynamic selection ration $\alpha \in [0, 1]$; three types of tokens IA, IC, and UIA (introduced in Section 3); cluster size $s$; decoding steps $E$.

**Output:** Evicted (Compressed) KV cache.

1  **Step 1: Look-once static eviction during prefill stage:**
2  **for** $l \in [0, \ldots, L-1]$ **do**
3  $\quad$ $\mathbf{Q}^{(l)}, \mathbf{K}^{(l)}, \mathbf{V}^{(l)} \leftarrow \mathcal{M}(\text{prompt})$;
4  $\quad$ **if** $l\%2 == 0$ **then**
5  $\quad\quad$ $\mathbf{ATTN}_{\text{window}}^{(l)} \leftarrow \text{Softmax}(\mathbf{Q}^{(l)}[:, -len(\text{prompt}) \times 0.2 :, :] \cdot \mathbf{K}^{(l)\top})$;
6  $\quad\quad$ $\{IA, IC, UIA\}^{(l)} \leftarrow \text{static\_evict}(\mathbf{ATTN}_{\text{window}}^{(l)})$;
7  $\quad$ **else**
8  $\quad\quad$ $\{IA, IC, UIA\}^{(l)} \leftarrow \{IA, IC, UIA\}^{(l-1)}$; $\qquad\qquad$ ▷ Layer index sharing (Section 4.4)
9  $\quad$ $\mathbf{K}^{(l)} \leftarrow \text{token\_gather}(\mathbf{K}^{(l)}, index = \{IA, IC\}^{(l)})$;
10 $\quad$ $\mathbf{V}^{(l)} \leftarrow \text{token\_gather}(\mathbf{V}^{(l)}, index = \{IA, IC\}^{(l)})$;
11 **Step 2: Query-aware dynamic selection during decoding stage:**
12 **for** $e \in [0, \ldots, E-1]$ **do**
13 $\quad$ **for** $l \in [0, \ldots, L-1]$ **do**
14 $\quad\quad$ **if** $l\%2 == 0$ **then**
15 $\quad\quad\quad$ $\mathbf{sim} \leftarrow \text{SimApprox}(\mathbf{q}^{(l,e)}, \mathbf{K}^{(l,e)}, cluster\_size = s)$; $\qquad$ ▷ Follow Equation (6)
16 $\quad\quad\quad$ $\mathbf{index}^{(l,e)} \leftarrow \text{topk}(\mathbf{sim}^{(l,e)}, k = len(\mathbf{K}^{(l,e)}) \times \alpha)$;
17 $\quad\quad$ **else**
18 $\quad\quad\quad$ $\mathbf{index}^{(l,e)} \leftarrow \mathbf{index}^{(l-1,e)}$; $\qquad\qquad\qquad\quad$ ▷ Layer index sharing (Section 4.4)
19 $\quad\quad$ $\mathbf{K}^{(l,e)} \leftarrow \text{token\_gather}(\mathbf{K}^{(l,e)}, index = \mathbf{index}^{(l,e)})$;
20 $\quad\quad$ $\mathbf{V}^{(l,e)} \leftarrow \text{token\_gather}(\mathbf{V}^{(l,e)}, index = \mathbf{index}^{(l,e)})$;
21 $\quad\quad$ $\mathbf{O}^{(l,e)} \leftarrow \text{Softmax}(\mathbf{q}^{(l,e)} \cdot \mathbf{K}^{(l,e)\top}/\sqrt{d}) \cdot \mathbf{V}^{(l,e)}$; $\qquad\qquad$ ▷ Sparse attention
22 **return** $\mathbf{K}, \mathbf{V}$.

---

## E  PARADIGM COMPARISON WITH DYNAMIC POLICY

We compare the paradigm between the dynamic algorithm and our MPCache as shown in Figure 16. Through performing static eviction during the prefill stage, we improve the efficiency of all decoding steps since the token selector (i.e., SimApprox and top-$k$ selection) only needs to handle a smaller number of tokens during the decoding stage (30% in this case). The improvement is shown in Section 5.4.

## F  SUPPLEMENTAL EXPERIMENTS

### F.1  SUPPLEMENTAL SETUPS

**Experimental envrionment.** The latency is evaluated under the LAN setup (Rathee et al., 2020) with 377MBps bandwidth and 0.3ms echo latency Rathee et al. (2020) on Intel(R) Xeon(R) Gold 5220R CPU @ 2.20GHz.

**Token clustering.** For hierarchy, we in practice choose a two-level hierarchical structure, i.e., $n = 2$, and when the final dynamic selection ratio $\alpha < 0.5$, we drop 50% clusters at the 1st hierarchical level. For the XSUM dataset, we use a cluster size of 8 at the 1st hierarchical level and 4 at the 2nd hierarchical level. For the long-context LongBench, we use larger clusters, i.e., 32 at the 1st hierarchical level and 16 at the 2nd hierarchical level.

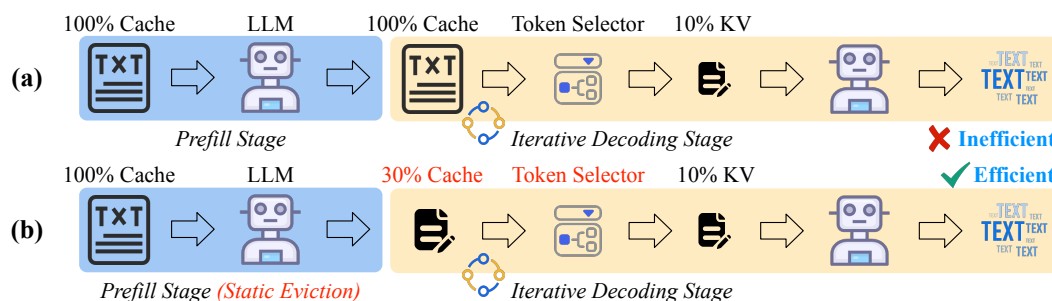

Figure 16: Paradigm comparison between (a) dynamic algorithm and (b) our proposed MP-Cache combining static eviction and dynamic selection. MPCache discards unimportant tokens to reduce the decoding overhead (red texts mean the differences).

**Static eviction.** During the static eviction, we compute the attention using the last 20% tokens in the prompt. However, using 20% tokens occurs CUDA out-of-memory (OOM) error when processing a long-context prompt (i.e., longer than 24k tokens). To solve this problem, we adaptively adjust to 10% tokens instead. We also notice that the choice of the ratio won't cause significant performance fluctuations, so we omit the discussion on the influence in this work.

## F.2  LLM ARCHITECTURES

We list the model architectures used in this work in Table 7, including GPT-2[3], LongChat-V1.5-7B[4], and LLaMA-3-8B[5]. LLaMA3 is equipped with grouped-query attention (GQA) (Ainslie et al., 2023) for improving the inference scalability.

Table 7: LLM architectures used in this work.

| Model | # Layers | # Attention Heads | # KV Heads | Embedding Dimension | Max Length | Non-linear Function | # Parameters |
|---|---|---|---|---|---|---|---|
| GPT-2 | 12 | 12 | 12 | 768 | 1024 | GeLU | 124M |
| LongChat-V1.5–7B | 32 | 32 | 32 | 4096 | 32K | SiLU | 7B |
| LLaMA-3-8B | 32 | 32 | 8 | 4096 | 8K | SiLU | 8B |

## F.3  DETAILED DESCRIPTION ABOUT BASELINES

StreamingLLM (Xiao et al., 2023) follows a fixed eviction pattern (keep local tokens and initial tokens) across different decoding steps. H2O (Zhang et al., 2024d) and TOVA (Oren et al., 2024) statically prune the KV cache and these discarded tokens cannot be recovered at subsequent decoding steps. SnapKV (Li et al., 2024) statically prunes the KV cache only during the prefill stage. InfLLM (Xiao et al., 2024) employs block-level dynamic token selection during the decoding stage. It requires selecting several representative tokens within a cluster and computing the relevance score using these representative tokens. LongCache (Liu et al., 2024b) uses the idea of cosine similarity between the query and key cache of all previous tokens to select relevant tokens without look-once static eviction and token clustering. Note that LongCache separates the positional embedding (PE) from the KV cache. Since our focus is on the dynamic selection metric in this work, we do not apply PE separation in LongCache.

## F.4  NECESSITY OF KV CACHE IN MPC

KV cache plays a crucial role in storing context information that the model deems relevant for subsequent token generation. KV cache eliminates the need for costly re-computation of previous key and value vectors (Ott, 2019). While the existing MPC framework PUMA (Dong et al., 2023) builds protocols to support private LLM inference, it still requires more than 40 seconds and 800

---

[3] https://huggingface.co/openai-community/gpt2
[4] https://huggingface.co/lmsys/longchat-13b-16k
[5] https://huggingface.co/meta-llama/Meta-Llama-3-8B-Instruct

MB to generate one token with a sequence length of 6 on GPT-2. The main reason is that it is not equipped with the KV cache, requiring re-computing the entire sequence for each token decoding. As shown in Table 8, KV cache for LLM generation is essential for private inference, bringing significant efficiency improvement, especially for longer sequences.

Table 8: Per-token generation efficiency on GPT-2 with and without KV cache.

| Method | Seq. Length=6 | | Seq. Length=16 | |
|---|---|---|---|---|
| | Lat. (s) | Comm. (GB) | Lat. (s) | Comm. (GB) |
| w/o KV cache | 19.78 | 0.273 | 44.40 | 0.872 |
| w/ KV cache | 7.890 | 0.068 | 8.313 | 0.071 |

### F.5 SUPPLEMENTAL ABLATION STUDY

**Effect of hyper-parameter $\alpha$.** To study how $\alpha$ impacts the similarity approximation, we select different $\alpha$'s on different datasets as shown in Figure 17. As can be observed, although the effects of different $\alpha$ do not occur in a certain pattern, we can still discover some patterns related to the dataset from the trend in Figure 17: on TriviaQA, the model may prefer larger $\alpha$ while it may prefer smaller $\alpha$ on HotpotQA instead. Since $\alpha = 0.6$ shows relatively better performance in these cases, we choose $\alpha = 0.6$ by default in our experiments.

**Comparison with average-based similarity approximation.** A straightforward and efficient way to aggregate the information of a key cache cluster is the average. We compare our proposed method ($\alpha = 0.6$) with average-based similarity on the XSUM dataset with a cluster size of 16 in Figure 17. Specifically, we perform dynamic selection with different ratios after 75% tokens are statically discarded. As can be observed, using average suffers from significant performance degradation under different ratios. With the compression ratio increasing, the degradation of the average-based method becomes more serious. An intuitive explanation is that using the average of a cluster may make some important tokens averaged and ignored. In contrast, our approximation can effectively maintain the model performance. We theoretically analyze the similarity approximation algorithm in Appendix G.

## G THEORETICAL ANALYSIS OF SIMILARITY APPROXIMATION

As mentioned in Section 2, given query $\mathbf{q} \in \mathbb{R}^{H \times 1 \times d}$, key cach $\mathbf{K} \in \mathbb{R}^{H \times T \times d}$, and value cache $\mathbf{V} \in \mathbb{R}^{H \times T \times d}$, the overall goal of KV cache eviction is to find an optimal policy $\mathcal{P}$ to minimize the gap between the attention outputs (here we omit $\mathbf{V}$ for simplification) as

$$\mathcal{P}^* = \arg\min |\text{Softmax}(\mathbf{q} \cdot \mathbf{K}^\top) - \text{Softmax}(\mathbf{q} \cdot \mathbf{K}'^\top)|, \quad (10)$$

where $\mathbf{K}'$ is a subset of $\mathbf{K}$ selected by the policy. However, when dividing $\mathbf{K}$ into clusters for efficiency, the problem becomes more challenging. We denote the key cache cluster as $\mathbf{K}_c$ (cluster

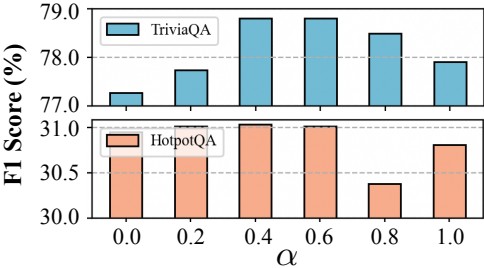

Figure 17: The influence trend of similarity approximation with different $\alpha$ values ranging from 0 to 1.

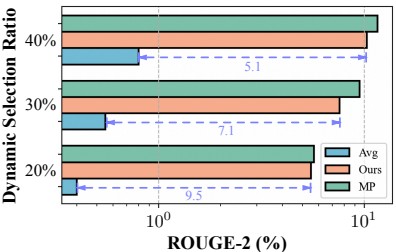

Figure 18: Comparison with average-based similarity approximation. MP means maximun dot product.

size is $s$), and the attention within the cluster can be computed as

$$\mathbf{A} = \mathrm{Softmax}(\mathbf{q} \cdot \mathbf{K}_c^\top) = \frac{\exp(\mathbf{q} \cdot \mathbf{K}_c^\top)}{\sum_{j=0}^{s-1} \exp(\mathbf{q} \cdot \mathbf{K}_{cj}^\top)}. \tag{11}$$

Our goal is to find a way to accurately approximate the similarity between $\mathbf{q}$ and the key cluster $\mathbf{K}_c$. Interestingly, this problem exists a dual problem: *"How can we aggregate the cluster information to obtain a cluster representation and measure its importance?"*

Note that $\exp(\mathbf{q} \cdot \mathbf{K}_c^\top)$ in the upper part of Equation (11) accurately computes the similarity between $\mathbf{q}$ and each token in $\mathbf{K}_c$, and our problem can be considered as approximating the lower part of Equation (11), i.e., $\sum_{j=0}^{s-1} \exp(\mathbf{q} \cdot \mathbf{K}_{cj}^\top)$.

We assume there exists a function $\phi$ that aggregates the key cluster $\mathbf{K}_c$, and we define the optimization problem as

$$\min |\sum_{j=0}^{s-1} \exp(\mathbf{q} \cdot \mathbf{K}_{cj}^\top) - \mathbf{q} \cdot \phi(\mathbf{K}_c^\top)|. \tag{12}$$

**Drawback of average-based clustering.** As mentioned in our main text, the simplest way to represent the cluster is the average and $\phi(\mathbf{K}_{cj}^\top)$ becomes $\sum_{j=0}^{s-1} \mathbf{K}_{cj}^\top / s$. This happens to be equivalent to directly drop $\exp$ in $\sum_{j=0}^{s-1} \exp(\mathbf{q} \cdot \mathbf{K}_{cj}^\top)$, introducing information loss. Intuitively, if there are tokens with very low importance and tokens with high importance within a cluster, the overall importance will be averaged, leading to the loss of crucial tokens.

Different from the average-based method, using the following max dot product can approximate the large values after $\exp$ more accurately and preserve the crucial tokens as much as possible. This observation is aligned with Tang et al. (2024b).

$$\mathrm{MaxDotProduct}: \quad \mathbf{q} \cdot \phi(\mathbf{K}_{cj}^\top) = \max_{\mathbf{k} \in \mathbf{K}_c} \mathbf{q} \cdot \mathbf{k}. \tag{13}$$

In order to approximate $\max_{\mathbf{k} \in \mathbf{K}_c} \mathbf{q} \cdot \mathbf{k}$ without accessing all the tokens in $\mathbf{K}_c$, we follow the bounding volume proposed by Klosowski et al. (1998) as described in Section 4.3. In fact, during the dynamic selection, $\mathbf{V}$ of a cluster can also influence the cluster's importance, and we will explore it in our future research.

## H   OVERALL SECURE INFERENCE FRAMEWORK

Take 2PC as an example in Figure 19, we illustrate the secure inference framework following Lu et al. (2023) where the server owns the proprietary LLM parameter and the client possesses private input data. During inference, the data is secretly shared between two parties. Linear layers are computed using the homomorphic encryption (HE) protocol, and non-linear layers require interactive protocols between the two parties based on oblivious transfer (OT) and HE. Figure 19 illustrates the detailed dataflow of MPCache during both the prefill and decoding stage. The static eviction algorithm is performed during the prefill stage (refer to Section 4.2) and the dynamic selection algorithm is performed during the decoding stage and before computing the attention, which relies on the min, max, matrix multiplication, top-$k$, and token gathering protocols (refer to Section 4.3).

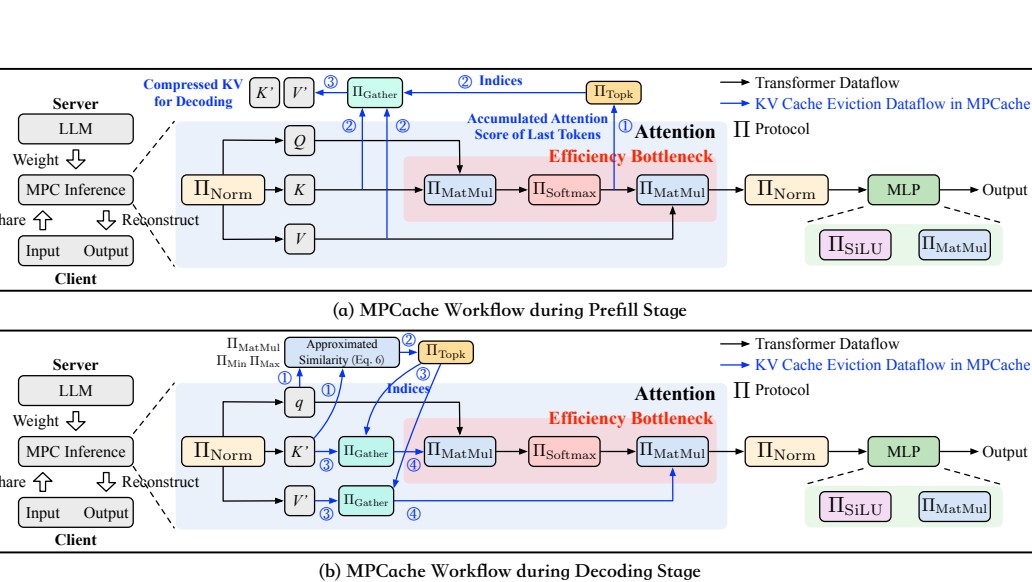

Figure 19: Overall secure inference framework and dataflow of MPCache during (a) the prefill stage and (b) the decoding stage.