# OpenReview forum: "MPCache: MPC-Friendly KV Cache Eviction for Efficient Private LLM Inference"
_ICLR.cc/2025/Conference — Submitted to ICLR 2025_

### Official Review · Reviewer_2Xee · 2024-10-28

**Soundness:** 2
**Presentation:** 3
**Contribution:** 1
**Rating:** 3
**Confidence:** 5

**Summary:**

This paper presents a solution to accelerate the Key Value cache eviction algorithms for the multi-party-based machine learning inference, so as to secure the inference of LLMs. The proposed method is built on the observation of unique features of long input sequences and dynamic KV cache selection.

**Strengths:**

This paper does statistical observation over the inference process of LLM models, especially their limits in advancing the inference overhead of a MPC-based LLM.

**Weaknesses:**

Threat Model: Applying privacy-preserving techniques to large language models (LLMs) is a valuable practice to enhance security and privacy. However, the practical setup is unclear, specifically regarding how secret-sharing is implemented across parties in two-party computation (2PC) or three-party computation (3PC) settings. For instance, could you clarify how the proposed method applies to 2PC, especially in the context of results shown in Figure 13?

Practicality: Current commercial LLMs are known for their substantial model sizes, raising questions about the feasibility of splitting them into different secret-shared components. The reviewer is uncertain whether multi-party computation (MPC) is a practical solution for LLMs under these conditions. Could you provide details on the setup that enables this approach?

Research Motivation: This work aims to integrate privacy-preserving machine learning as a service (MLaaS) with key-value (KV) usage to improve ML inference. However, the reviewer notes that the primary bottleneck in existing privacy-preserving ML research is typically focused on improving efficiency, particularly the low performance associated with non-linear operators. Could you clarify how the Key-Value (KV) approach addresses this performance issue, which is crucial to the MLaaS context? The reviewer believes prioritizing efficiency improvements in MLaaS performance would be more impactful than integrating additional components that excel in their own specialized areas.

Experimental Results: The experimental setup and results provide useful insights into the performance of the proposed solutions. However, the reviewer suggests that these results should be compared to existing work within privacy-preserving MLaaS, rather than solely with other KV cache eviction approaches. The use of KV in this context is intended to enhance MLaaS inference efficiency in 2PC and 3PC settings, where numerous state-of-the-art (SOTA) solutions exist. Therefore, comparing MPCache's performance to other non-KV-based solutions would offer a clearer benchmark, as they represent the current SOTA in this domain.

**Questions:**

My questions are embedded in the weakness section, please refer to them.

---

> ### Author Response · Authors · 2024-11-20
> **Response to Reviewer 2Xee (Part 1/2)**
>
> We sincerely thank Reviewer 2Xee for your thoughtful feedback! Below we answer each of your comments and questions:
>
> **Question 1:** Practical setup is unclear. How secret-sharing is implemented across parties in two-party computation (2PC) or three-party computation (3PC) settings.
>
> **Response 1:**
> Thanks for your comments on the MPC settings. Here, we provide more details about the 2PC and 3PC settings.
> - For 2PC setting, we follow the 2PC protocol BumbleBee where the server owns the proprietary LLM parameter and the client possesses private input data. During 2PC inference, the data is secretly shared between two parites. The linear layers are computed using homomorphic encryption (HE) and non-linear layers require interactive protocols between the two parties based on oblivious transfer (OT) and HE.
> - For 3PC setting, we follow the 3PC protocol PUMA where the parameter and data are both secretly shared among three parties. During the 3PC inference process, the linear layers are computed using 2-out-of-3 replicated secret sharing (RSS) and non-linear layers are computed using OT.
>
> We describe the threat model in Appendix B and also add an overall secure inference framework in Figure 19 in Appendix H (refer to the updated Supplementary Material). The static eviction algorithm is performed during the prefill stage (refer to Section 4.2), and the dynamic selection algorithm is performed during the decoding stage and before computing the attention, which relies on the matrix multiplication, top-k, and token gathering protocols (refer to Section 4.3). All protocols in MPCache operate on ciphertext. Since our focus is on the algorithm optimization rather than protocol optimization, we put more information to the appendix, and the security of MPCache directly follows the security from PUMA and BumbleBee. We will add more detailed practical setups in our final version.
>
> ---
>
> **Question 2:** Uncertain whether MPC is a practical solution for LLMs.
>
> **Response 2:**
> Although MPC iccurs high communication and latency overhead, we believe MPC has strong potential for the following reasons:
> - MPC provides cryptographically-strong privacy protection for both data and the LLM parameters and is also independent of the hardware platforms. Compared with HE-based inference methods, MPC inference is more accurate.
> - In recent years, there is a significant amount of research on MPC inference: 1) MPC-friendly algorithm optimization [1-6]. For example, MPCFormer [2] achieves up to 5x speedup without comprimising the accuracy; 2) MPC protocol optimization [7-10]. For example, PUMA [8] achieves up to 3x speedup compared to MPCFormer;  3) hardware acceleration [11-13]. For example, [12] can achieve more than 100x speedup for the linear layers, which is computed with HE in 2PC scenario. These optimizations are orthogonal to each other and thus can be combined together to advance the feasibility of MPC inference.
>
> While MPC for LLMs still incurrs high overhead, our work attempts to optimize the efficiency from the algorithm perspective. Specifically, MPCache considers the cost of MPC protocols to design an MPC-friendly algorithm, aiming to further improve efficiency and promote its application. More importantly, our work does not rely on specific MPC protocols and can be naturally extended to other schemes besides PUMA and BumbleBee. Our method is also orthogonal to other algorithm/protocol/hardware optimizations, and they can be combined togher to further improve the efficiency.
>
> ---
>
> **Question 3:** Motivation of this work.
>
> **Response 3:**
> Our work also focuses on improving the efficiency of private inference, just like existing privacy-preserving ML research.
> We want to clarify that the main contribution of our work is not combining MLaaS with KV cache usage. Instead, KV cache is already a fundamental component for existing generative LLMs to guarantee the inference correctness. If we do not use KV cache, the model has to re-compute all the previous tokens, significantly increasing the overhead (details can be found in Appendix A.3). Hence, we apply KV cache to private LLM inference by default, and our overall aim is to reduce the computation associated with KV cache size for better efficiency.
> We find the communicaiton and latency overhead mainly come from attention computation with KV cache, especially non-linear Softmax as shown in Figure 1(a).
> Hence, MPCache develops an MPC-friendly KV cache eviction algorithm to significantly reduce the overhead of expensive non-linear Softmax.

---

> ### Author Response · Authors · 2024-11-20
> **Response to Reviewer 2Xee (Part 2/2)**
>
> **Question 4:** Compared to existing work within privacy-preserving MLaaS.
>
> **Response 4:**
> In recent years, a large number of algorithm optimization solutions for efficient MPC inference have emerged, such as non-linear ReLU and GeLU pruning/approximation [2,3,4,6], Softmax approximation [2,3,5], etc. However, directly applying these methods to LLMs seriously damages the model accuracy. For example, directly replacing exponential in Softmax with ReLU following MPCViT [3] drastically degrades the accuracy from 78.01% to 25.04% on LLaMA-3-8B and HellaSwag dataset. Hence, these methods inevitably require expensive retraining or finetuning to recover the model accuracy, which is unrealistic and still remains unclear for LLMs.
> In this work, we propose MPCache, the first framework that leverages KV cache compression technique for private LLM inference. MPCache is based on post training and totally avoids the expensive cost of training or finetuning.
> In our experimens, we have applied our method to both SOTA 3PC protocol PUMA and 2PC protocol BumbleBee. As a result, our work can achieve more than 8x communication reduction with a sequence length of 2048 using PUMA.
>
> ---
>
> **References:**
>
> [1] Rathee, Deevashwer, et al. "MPC-Minimized Secure LLM Inference." arXiv preprint arXiv:2408.03561 (2024).
>
> [2] Li, Dacheng, et al. "Mpcformer: fast, performant and private transformer inference with mpc." ICLR, 2023.
>
> [3] Zeng, Wenxuan, et al. "Mpcvit: Searching for accurate and efficient mpc-friendly vision transformer with heterogeneous attention." Proceedings of the IEEE/CVF International Conference on Computer Vision. 2023.
>
> [4] Zhang, Yuke, et al. "Sal-vit: Towards latency efficient private inference on vit using selective attention search with a learnable softmax approximation." Proceedings of the IEEE/CVF International Conference on Computer Vision. 2023.
>
> [5] Li, Fabing, et al. "Seesaw: Compensating for Nonlinear Reduction with Linear Computations for Private Inference." ICML 2024.
>
> [6] Kundu, Souvik, et al. "Learning to linearize deep neural networks for secure and efficient private inference." ICLR 2023.
>
> [7] Zheng, Fei, et al. "PermLLM: Private Inference of Large Language Models within 3 Seconds under WAN." arXiv preprint arXiv:2405.18744 (2024).
>
> [8] Dong, Ye, et al. "Puma: Secure inference of llama-7b in five minutes." arXiv preprint arXiv:2307.12533 (2023).
>
> [9] Lu, Wen-jie, et al. "Bumblebee: Secure two-party inference framework for large transformers." Cryptology ePrint Archive (2023).
>
> [10] Pang, Qi, et al. "Bolt: Privacy-preserving, accurate and efficient inference for transformers." 2024 IEEE Symposium on Security and Privacy (SP). IEEE, 2024.
>
> [11] Jiang, Wuxuan, et al. "Spin: An Efficient Secure Computation Framework with GPU Acceleration." arXiv preprint arXiv:2402.02320 (2024).
>
> [12] Fan, Shengyu, et al. "Tensorfhe: Achieving practical computation on encrypted data using gpgpu." 2023 IEEE International Symposium on High-Performance Computer Architecture (HPCA). IEEE, 2023.
>
> [13] Samardzic, Nikola, et al. "Craterlake: a hardware accelerator for efficient unbounded computation on encrypted data." Proceedings of the 49th Annual International Symposium on Computer Architecture. 2022.

---

> ### Author Response · Authors · 2024-11-23
> **Follow-up on Rebuttal for Paper 10561 - Request for Response**
>
> Dear Reviewer 2Xee,
>
> We are writing to follow up on the rebuttal submission for our paper. We have carefully addressed all the concerns you raised in your initial review, providing additional clarifications to support our responses.
>
> Given the significant impact of your review on the final decision, we believe our rebuttal must be fully considered. We kindly request your prompt feedback on our responses, as your feedback is essential to ensuring a fair and balanced evaluation of our work.
>
> Please let us know if there are any additional questions you would like us to address or if further clarification is required.
>
> Thank you for your attention to this matter!

---

> > ### Comment · Reviewer_2Xee · 2024-11-23
> > **Thank you for the detailed responses, which have clarified some of my earlier questions.**
> >
> > However, I still have concerns regarding the research motivation and methodology. Specifically, two critical issues remain:
> >
> > 1. I find it impractical, if not impossible, to apply MPC to LLMs given their ultra-large size. This approach appears more like a combination of two buzzwords rather than addressing any real-world needs for current LLMs, especially considering their already commercialized state.
> >
> > 2. Similarly, the integration of KV mechanisms seems to follow the same keywords-driven approach. As highlighted in the first concern, the intensive computational workload of LLMs limits the feasibility of adopting MPC. I do not see how the inclusion of KV mechanisms contributes to addressing this issue, nor do I find a compelling necessity for it in this context.

---

> > > ### Author Response · Authors · 2024-11-26
> > > **Thank you very much for your feedback - Response to Reviewer 2Xee**
> > >
> > > We sincerely thank Reviewer 2Xee for your feedback. Below we answer each of your concerns:
> > >
> > > ---
> > >
> > > **Question 1:** Applying MPC to ultra-large LLMs is impractical.
> > >
> > > **Response 1:**
> > > Although MPC incurs high overhead and is still under research, we believe that MPC has a very strong potential as we mentioned above. First, MPC can provide cryptographically-strong privacy protection for both private data and the proprietary model parameters, and MPC has been widely deployed to various models such as convolutional neural networks (CNN), vision transformer (ViT), gradient boosting decision tree (GBDT), etc [1-5].
> > >
> > > For LLMs, the issue of privacy protection becomes more severe, and there is a much stronger demand for LLM privacy protection. In recent years, applying MPC to LLM is a very important topic, which has been widely researched in both academia and industry to address the privacy concern [6-11]. However, there is still a lack of highly-efficient solutions for LLM privacy at present. In our work, we attempt to optimize the efficiency bottleneck, and we take a step forward in the feasibility of MPC-based LLM inference.
> > > As observed in our experiments, we have evaluated LLMs with 7B parameters, so we believe that applying MPC to LLM is a promising solution.
> > >
> > > Moreover, we are not simply combining MPC and LLM, but have designed a novel algorithm that carefully considers the characteristics of MPC. Our proposed techniques, e.g., approximated top-k algorithm, can be also applied to other applications such as vector database and retrieval-augmented generation (RAG). Therefore, we believe our research has good universality and important academic value.
> > >
> > > The reviewer simply thinks that applying MPC to LLM is impractical and negates the academic significance of our work. Therefore, we strongly urge the reviewer to re-assess our work.
> > >
> > > In conclusion, we believe MPC for LLM is an important topic, and our key motivation is to improve the private inference efficiency. Also, we would like to know if the reviewer have any technical suggestions for our work.
> > >
> > > ---
> > >
> > > **Question 2:** How the inclusion of KV mechanisms contributes to addressing the intensive computational workload of LLMs.
> > >
> > > **Response 2:**
> > > We want to clarify that KV cache is already a fundamental technique in existing LLMs [12-14], which stores the past computed KV cache to avoid the heavy re-computation of the previous tokens as well as the intensive computational workload of LLMs (we describe the details in Appendix A.3).
> > > For private LLM inference, using KV cache is also very important for efficiency by avoiding the heavy re-computation of the previous tokens during inference (results are shown in experiments in Figure 10(left purple bars) and Appendix F.4).
> > > Hence, we apply KV cache to LLM inference by default, and our overall motivation/aim is to reduce the computation by reducing the KV cache size, i.e., token number for better inference efficiency.
> > >
> > > ---
> > >
> > > **References:**
> > >
> > > [1] Li, Dacheng, et al. "Mpcformer: fast, performant and private transformer inference with mpc." ICLR. 2023.
> > >
> > > [2] Zeng, Wenxuan, et al. "Mpcvit: Searching for accurate and efficient mpc-friendly vision transformer with heterogeneous attention." Proceedings of the IEEE/CVF International Conference on Computer Vision. 2023.
> > >
> > > [3] Ma, Junming, et al. "SecretFlow-SPU: A Performant and User-Friendly Framework for Privacy-Preserving Machine Learning." 2023 USENIX Annual Technical Conference (USENIX ATC 23). 2023.
> > >
> > > [4] Lu, Wen-jie, et al. "Squirrel: A Scalable Secure {Two-Party} Computation Framework for Training Gradient Boosting Decision Tree." 32nd USENIX Security Symposium (USENIX Security 23). 2023.
> > >
> > > [5] Rathee, Deevashwer, et al. "Sirnn: A math library for secure rnn inference." 2021 IEEE Symposium on Security and Privacy (SP). IEEE, 2021.
> > >
> > > [6] Rathee, Deevashwer, et al. "MPC-Minimized Secure LLM Inference." arXiv preprint arXiv:2408.03561 (2024).
> > >
> > > [7] Zheng, Fei, et al. "PermLLM: Private Inference of Large Language Models within 3 Seconds under WAN." arXiv preprint arXiv:2405.18744 (2024).
> > >
> > > [8] Pang, Qi, et al. "Bolt: Privacy-preserving, accurate and efficient inference for transformers." 2024 IEEE Symposium on Security and Privacy (SP). IEEE, 2024.
> > >
> > > [9] Hou, Xiaoyang, et al. "Ciphergpt: Secure two-party gpt inference." Cryptology ePrint Archive (2023).
> > >
> > > [10] Dong, Ye, et al. "Puma: Secure inference of llama-7b in five minutes." arXiv preprint arXiv:2307.12533 (2023).
> > >
> > > [11] Lu, Wen-jie, et al. "Bumblebee: Secure two-party inference framework for large transformers." Cryptology ePrint Archive (2023).
> > >
> > > [12] Brown, Tom B. "Language models are few-shot learners." arXiv preprint arXiv:2005.14165 (2020).
> > >
> > > [13] Touvron, Hugo, et al. "Llama: Open and efficient foundation language models." arXiv preprint arXiv:2302.13971 (2023).
> > >
> > > [14] Zhang, Susan, et al. "Opt: Open pre-trained transformer language models." arXiv preprint arXiv:2205.01068 (2022).

---

### Official Review · Reviewer_h8w6 · 2024-10-30

**Soundness:** 3
**Presentation:** 4
**Contribution:** 3
**Rating:** 8
**Confidence:** 3

**Summary:**

The paper introduces a new design for the KV cache in large language models (LLMs) aimed at enhancing compatibility with MPC (multi-party computation) frameworks. It highlights that the existing KV cache design incurs significant latency and communication overhead, making it inefficient for MPC applications.

To address these inefficiencies, the authors propose three main optimizations:
1. Attention-based KV cache eviction: This approach prioritizes efficiency by evicting cache entries based on attention scores.
2. Dynamic KV cache selection algorithm: Optimized to remove operations unsuitable for MPC, this algorithm improves MPC compatibility.
3. Layer-wise index sharing strategy: By sharing indices across layers, this strategy reduces the number of tokens stored, further minimizing overhead.

**Strengths:**

The paper presents the underlying questions and motivations, offering a clear explanation of the challenges addressed. It includes ample experimental results that demonstrate how the proposed optimizations improve overhead and latency. The structure of the paper makes it easy to follow.

Additionally, the attached appendix provides extensive information and additional experimental results, enriching the reader’s understanding of the work and its impact.

**Weaknesses:**

I didn’t identify any major weaknesses in this paper. It is well-structured, with sufficient explanations and experimental results to support its findings. Although the main content may be concise, likely due to page limitations, this doesn’t detract from the paper, as the appendix compensates by providing additional, valuable information.

**Questions:**

Thanks for this contribution. I have 2 questions:

Question 1: In Section 4.3, Hierarchical KV Cache Clustering, could you specify the number of levels (n) selected for clustering in your experiments? Additionally, Figure 6 would benefit from further explanation—could you expand on the section?

Question 2: In Figure 8, I noticed a lower commonality score between layers 6 and 12. Could you explain what might cause this dip? Also, is there any observed performance degradation when applying the sharing strategy in these layers? Any insights into this effect would be helpful.

---

> ### Author Response · Authors · 2024-11-20
> **Response to Reviewer h8w6**
>
> We are very grateful for the Reviewer h8w6’s appreciation of our work and the thoughtful feedback! Below we answer each of your concerns:
>
> ---
>
> **Question 1:** Specify the number of levels (n) selected for clustering in the experiments.
>
> **Response 1:**
> For hierarchical clustering, we in practice choose a two-level hierarchical structure, i.e., n=2, and when the final dynamic selection ratio < 0.5, we first drop 50% clusters at the 1st hierarchical level. In our experiments on the long-context LongBench, we choose to use a cluster size of 32 at the 1st hierarchical level and 16 at the 2nd hierarchical level to trade off the efficiency and model performance. The detailed experimental settings of the hierarchical clustering can be found in Appendix F, and we will describe more details about Figure 6 in our final version.
>
> ---
>
> **Question 2:** Lower commonality score between layers 6 and 12.
>
> **Response 2:**
> Since the middle layers also show a little lower commonality score, we compare the performance with and without applying the layer-wise sharing strategy in the middle layers under different KV cache budgets in the following table. As shown in the table, applying layer-wise sharing strategy to these layers have acceptable influence on the model performance.
>
> | KV Cache Budget | w/o Sharing Middle Layers | w/ Sharing Middle Layers |
> | --- | --- | --- |
> | 3% | 29.95 | 29.55 |
> | 6% | 30.53 | 30.27 |
> | 12% | 30.37 | 30.00 |
>
> We also think this is an interesting phenomenon and we will further explore this problem in our future work.

---

> > ### Comment · Reviewer_h8w6 · 2024-11-27
> >
> > Thank you for thoughtfully addressing my concerns. I truly appreciate your efforts and am excited to explore your future work!

---

### Official Review · Reviewer_782C · 2024-11-03

**Soundness:** 2
**Presentation:** 3
**Contribution:** 3
**Rating:** 3
**Confidence:** 4

**Summary:**

In this paper, the authors introduce an MPC-friendly KV cache eviction framework, which accelerates the private LLM inference with secure KV cache eviction. The authors first incorporate the look-once static eviction algorithm and the query-aware dynamic selection algorithm to reduce the number of tokens needed for inference. To improve the efficiency of the dynamic selection algorithm, the authors propose several strategies such as MPC-friendly similarity approximation, hierarchical KV cache clustering, and layer-wise index sharing. The experiments show that the proposed methods can improve the latency by 2.01 times and communication by 8.37 times.

**Strengths:**

1. Protecting user privacy during LLM inference is important. Integrating static and dynamic KV cache eviction methods is a promising direction for improving the efficiency of private LLM inference.
2. The authors present comprehensive experimental results to support their motivation and to show the effectiveness of the proposed methods.
3. Although some technical details can benefit from clearer explanation, the manuscript is overall well-structured and easy to follow.

**Weaknesses:**

1. Key technical issue. The secure token gathering protocol is the core of MPCache, as MPCache focuses on identifying and removing less important tokens during private LLM inference. However, the proposed token gathering protocol is problematic. While the top-k indices can be turned into a one-hot vector via the secure comparison protocol, directly multiplying the cache tokens with the one-hot vector cannot remove tokens. Instead, parties will learn a token sequence of the same length after running the proposed token gathering protocol. Additionally, no party will know which token should be removed since all tokens are secretly shared. The proposed token gathering protocol is not sound and fails to implement the expected functionality shown in Fig. 14. Since the token gathering protocol is needed for almost all experiments, I am concerned about the soundness of this paper.
2. There is neither a detailed MPC protocol describing the whole secure inference process in MPCache, nor an implementation of the MPC backend in the provided code. Additionally, while the authors argue that MPCache is built upon existing cryptographic primitives, more information is revealed to the parties (e.g., how many tokens in the cache are used). A security analysis or proof is still needed.
3. There are multiple unclear citations. For example, in line 52, the author cited Autorep (ICCV'23) for "replacing non-linear activation functions with more MPC-friendly operators" to "reduce the cost of private LLM inference". Yet, Autorep is not about private LLM inference. In line 592, the author cited Lazyllm, but it does not appear in the main text. While I am aware that these papers are related, clearer citations can enhance the overall presentation.
4. As shown in Fig. 9, MPCache presents better performance than most KV cache eviction methods in the plaintext (both static and dynamic ones). Such results seem to make MPCache a competitive work in plaintext. However, the gains in secure computation do not mainly come from the proposed optimizations. As shown in Fig. 11, the improvement in latency can be largely attributed to the top-k parallel (line 412) and the communication due to the static eviction. The main techniques in Section 4.3 and 4.4 contribute to the end-to-end performance much more marginally when compared with top-k parallel and static eviction.

**Questions:**

1. The proposed MPCache involves multiple hyperparameters such as $\alpha$ (line 340), $\gamma$ (line 287), the number of adjacent layers affected, and the eviction ratio. How should these parameters be determined in practice? Should they be determined by the client?
2. MPCache mainly focuses on the eviction of the key cache. How does the eviction of the value cache influence the overall performance?
3. How does the parallel top-k protocol differ from that in existing works such as CipherGPT?

---

> ### Author Response · Authors · 2024-11-20
> **Response to Reviewer 782C (Part 1/2)**
>
> We sincerely thank Reviewer 782C for your thoughtful feedback! Below we answer each of your comments and questions:
>
> ---
>
> **Question 1:** Directly multiplying the cache tokens with the one-hot vector cannot remove tokens.
>
> **Response 1:**
> Sorry for the confusion caused by our unclear description. Here we give a more clear explanation. We first take retrieving 1 token as an example. The first step of token gathering is converting an index $i$ into an one-hot vector $o\in\mathbb R^{1\times T}$ based on comparison protocol, where $T$ denotes token number. Given a key cache $K\in\mathbb R^{T\times D}$, where $D$ denotes the hidden dimension, multiplying $o$ with $K$ (matrix-vector multiplication) can generate an output with dimension $1\times D$, which is the retrieved token. To extend the case to retrieve $m$ tokens, we concatenate $m$ one-hot vectors to form a matrix $O\in \mathbb R^{m\times T}$, and then multiply $O$ with $K$ (matrix-matrix multiplication) to generate an output with dimension $m\times D$, which is the retrieved tokens. In our updated Supplementary Material, we have improved the protocol description in Appendix B.4.
>
> ---
>
> **Question 2:** Paper lacks description of the whole secure inference process, and more information is revealed (e.g., how many tokens in the cache are used).
>
> **Response 2:**
> Thanks for your kind advice! We add an overall framework of the whole secure inference process as shown in Figure 19 in Appendix H (refer to the updated Supplementary Material), and we will add it to our final version. Take 2PC inference as an example as illustrated in Figure 19, the server owns the proprietary LLM parameter and the client possesses private input data. During 2PC inference, the data is secretly shared between two parites. Following BumbleBee, linear layers are computed using homomorphic encryption (HE) and non-linear layers require interactive protocols between the two parties based on oblivious transfer (OT) and HE.
> The static eviction algorithm is performed during the prefill stage (refer to Section 4.2), and the dynamic selection algorithm is performed during the decoding stage and before computing the attention, which relies on the matrix multiplication, top-k, and token gathering protocols (refer to Section 4.3).
>
> MPCache follows a common threat model where all the parties are semi-honest, i.e., the parties follow the designed protocols but are curious and attempt to learn extra information. In our threat model, we assume all the parties are aware of the LLM architecture and eviction ratios, which is consistent with HEPrune [1], Seesaw [2], SENet [3], SNL [4], etc. We argue that this information does not compromise the client’s data or inference results, nor does it enable the client to access the model’s parameters.
>
> In practice, we believe the static and dynamic eviction ratio should be determined based on two aspects: 1) the expected sequence length and 2) the latency that can be tolerated/afforded by the server and the client.
> Hence, eviction ratio does not reveal the information of data and model parameters.
> Our framework proposes a novel algorithm with configurable hyper-parameters to enable exploring the Pareto front of the LLM performance and efficiency.
>
> ---
>
> **Question 3:** Unclear citations.
>
> **Response 3:**
> Thanks for your kind suggestion! We have revised the unclear citations in our updated version, and we will organize the citations more carefully to enhance the paper presentation in our final version.

---

> ### Author Response · Authors · 2024-11-20
> **Response to Reviewer 782C (Part 2/2)**
>
> **Question 4:** The main techniques in Section 4.3 and 4.4 contribute to the end-to-end performance more marginally when compared with top-k parallel and static eviction.
>
> **Response 4:**
> We agree that top-k and static eviction contribute most in our ablation study. However, for smaller models, e.g., GPT-2, the overhead proportion of the max protocol and multiplication in Equation (4) becomes larger. Below we give example cases on GPT-2 with a sequence length of 512 and 1024 to show the effectiveness of our proposed techniques. As observed, techniques, i.e., Linearization & Reordering (Section 4.3) and Layer-wise Index Sharing (Section 4.4) both contribute to the efficiency improvement significantly.
>
> | Seq. Len=512 | Dynamic Selection w/o Opt.  | + Top-k Parallel | + Layer-wise Index Sharing | + Linearization & Reordering |
> | --- | --- | --- | --- | --- |
> | Comm. (MB)  | 104.58 | 104.58  | 74.82  |  53.49 |
> | Improve   | 1.00x | 1.00x | 1.40x | 1.96x |
> | Lat. (s)  | 12.240 | 8.370 | 6.545  | 4.446 |
> | Improve   | 1.00x | 1.46x | 1.87x | 2.75x |
>
> | Seq. Len=1024 | Dynamic Selection w/o Opt.  | + Top-k Parallel | + Layer-wise Index Sharing | + Linearization & Reordering |
> | --- | --- | --- | --- | --- |
> | Comm. (MB)  | 169.4 | 169.4  | 109.79  | 67.04  |
> | Improve   | 1.00x | 1.00x | 1.54x | 2.53x |
> | Lat. (s)  | 18.823 | 12.976 | 9.848  | 7.914 |
> | Improve   | 1.00x | 1.45x | 1.91x | 2.38x |
>
> In conclusion, the techniques proposed in Section 4.3 and 4.4 are orthogonal to other optimizations, and they are all MPC-friendly and also have different impact in different situations. We will add more comprehensive experimental results and analysis in our final version.
>
> ---
>
> **Question 5:** How should parameters (alpha, gamma, #adjacent layers, eviction ratio) be determined in practice.
>
> **Response 5:**
> In practice, these parameters are pre-determined before inference and public to all the parties. In our experiments, $\alpha$ used in similarity approximation is empirically determined and we fix it to 0.6 (ablation study in Figure 17). The number of adjacent layers for layer-wise index sharing is fixed to 2 for better accuracy (ablation study in Figure 12). In practice, we believe the eviction ratio should be determined based on two aspects: 1) the expected sequence length and 2) the latency that can be tolerated/afforded by the server and the client. MPCache proposes a novel algorithm with configurable hyper-parameters to enable exploring the Pareto front of the LLM performance and efficiency.
>
> **Question 6:** How does the eviction of the value cache influence the overall performance.
>
> **Response 6:**
> For KV cache eviction, once key cache is pruned, the corresponding value cache is useless. Hence, key cache and value cache are simultaneously pruned correspondingly based on the selected token indices, as described in Algorithm 1. In our experiments, we prune both key and value cache.
>
> ---
>
> **Question 7:** The difference between parallel top-k protocol and existing works such as CipherGPT.
>
> **Response 7:**
> We follow a similar top-k protocol as CipherGPT. However, the top-k protocol in CipherGPT is designed for selecting k elements from a single vector and not optimized with parallelization, and hence, for token selection in this work, multiple attention heads need to be computed sequentially, leading to significant overhead. In our work, we improve the top-k protocol with parallelization along the attention head axis to support the head-wise token selection, which is much more efficient than the top-k protocol in CipherGPT.
>
> ---
>
> **References:**
>
> [1] Zhang, Yancheng, et al. "HEPrune: Fast Private Training of Deep Neural Networks With Encrypted Data Pruning." NeurIPS 2024.
>
> [2] Li, Fabing, et al. "Seesaw: Compensating for Nonlinear Reduction with Linear Computations for Private Inference." ICML 2024.
>
> [3] Kundu, Souvik, et al. "Learning to linearize deep neural networks for secure and efficient private inference." ICLR 2023.
>
> [4] Cho, Minsu, et al. "Selective network linearization for efficient private inference." ICML 2022.

---

> ### Author Response · Authors · 2024-11-23
> **Follow-up on Rebuttal for Paper 10561 - Request for Response**
>
> Dear Reviewer 782C,
>
> We are writing to follow up on the rebuttal submission for our paper. We have carefully addressed all the concerns you raised in your initial review, providing additional data and clarifications to support our responses.
>
> Given the significant impact of your review on the final decision, we believe our rebuttal must be fully considered. We kindly request your prompt feedback on our responses, as your feedback is essential to ensuring a fair and balanced evaluation of our work.
>
> Please let us know if there are any additional questions you would like us to address or if further clarification is required.
>
> Thank you for your attention to this matter!

---

> ### Comment · Reviewer_782C · 2024-11-23
>
> Thanks to the authors for their detailed response.
>
> Many of my concerns are addressed. The updated description about the token gathering protocol for one-token retrieving in Appendix B.4 and Algorithm 2 is well-organized to me. Additionally, I encourage the authors to include the discussion about the security threats as well as hyperparameter setting to your final version. It is challenging for the client and server to pre-determine the hyperparameter before inference. The client has limited knowledge about the proprietary model of the server, it would be difficult for the client to see the impact of different choices of parameters like $\alpha$ and #adjacent layers. The pre-defined parameters such as $\alpha=0.6$ and #adjacent layers $=2$ might not be robust for inputs of different lengths and from different datasets. I believe acknowledging this challenge can help the readers to better understand how MPCache can be deployed in practice.
>
> However, some of my concerns remain unsolved.
>
> * The protocols in MPCache. I understand that MPCache involves the MatMult, TopK and Gather protocols as shown in Figure 19. However, I have concerns about how these protocols are used in MPCache.
>     * For the Gather protocol, to generate the one-hot vector, equality protocol should be used instead of comparison (Algorithm 2 line 2). Comparison protocols returns 0 if the $id$ is smaller than $i$ and 1 otherwise (or vice versa, as shown in Figure 3 in Iron).
>     * For the TopK protocol, the authors mentioned that they follow the design of CipherGPT, except for the parallelization across the attention heads. The TopK protocol in CipherGPT is shuffle-based, because the comparison results need to be revealed for efficient quick sorting (Algorithm 2 line 8 in CipherGPT). With the shuffle-based TopK protocol, the relative order of tokens and KV caches cannot be preserved. However, as shown in Appendix D of Bolt (S&P 2024), the relative order of tokens is important during pruning.
>
> * The secure computation framework is still unclear. The standard secure inference process of Transformers, such as that in Iron and BumbleBee, is clear. However, how MPCache is integrated in this process still remains unclear. Figure 19 does not correctly reflect the workflow of MPCache (in Algorithm 3). Since the KV Eviction is performed before attention in Figure 19, I assume that Figure 19 only illustrates the decoding step and the dynamic selection. There are multiple issues. For example, the query $q$ is needed to calculate the similarity, but is not passed to the KV Eviction module. Also, $Q$ is used to denote the queries in the prefilling phase. The static eviction is performed after the attention, which is not reflected in Figure 19 or Appendix H.
> * I understand that both key cache and value cache are evicted (line 19 and 20 in Algorithm 3). During dynamic selection, if the similarity is calculated using the value cache, will the final performance be the same?
>
> There are still some minor issues about the citation. For example, there are two citations for LazzyLLM (line 594 and 598). I encourage the authors to check their cited papers to make sure the references are correct.

---

> > ### Author Response · Authors · 2024-11-25
> > **Thank you very much for your helpful feedback - Response to Reviewer 782C**
> >
> > We sincerely thank Reviewer 782C for your kind feedback! Below we answer each of your comments:
> >
> > ---
> >
> > **Question 1:** Include the discussion about the security threats and hyperparameter setting to the final version.
> >
> > **Response 1:** Thanks for your valuable advice! We will add the detailed discussion about the security threats and hyperparameters in our final version, and carefully revise our writing.
> >
> > ---
> >
> > **Question 2:** Gathering protocol uses equality protocol instead of comparison in Algorithm 2 line 2.
> >
> > **Response 2:** Sorry for our inaccurate description. It should be the equal protocol, and we have revised the writing in our updated PDF.
> >
> > ---
> >
> > **Question 3:** With the shuffle-based top-k protocol, the relative order of tokens and KV caches cannot be preserved.
> >
> > **Response 3:** We agree that the relative order of tokens cannot be preserved with shuffle-based top-k during pruning, but we believe that it
> > won't impact the result of attention computation.
> > This is because:
> > 1) Key and value cache are simultaneously pruned based on the top-k indices.
> > 2) Each token are embedded with positional embedding already.
> >
> > Therefore, as long as the correct top-k indices are selected, the result should remain the same.
> >
> > Here, we give an intuitive example with top-3 tokens to explain the computation correctness. Denote $d$ as the hidden dimension, $q\in\mathbb R^{1\times d}$ as the query, $K\in\mathbb R^{3\times d}$ as the key cache, and $V\in\mathbb R^{3\times d}$ as the value cache. $K$ and $V$ can be re-written as
> > $
> > K=\begin{bmatrix}
> > k_1 \\
> > k_2 \\
> > k_3 \\
> > \end{bmatrix},
> > V=\begin{bmatrix}
> > v_1 \\
> > v_2 \\
> > v_3 \\
> > \end{bmatrix}
> > $
> > , where $k_i\in\mathbb R^{1\times d}, v_i\in\mathbb R^{1\times d}$ denotes the $i$-th token in key and value cache.
> > Then, the attention $\mathrm{Softmax}(q\cdot K^T)\cdot V$ can be computed as
> > $$
> > \mathrm{Softmax}(q\cdot
> > \begin{bmatrix}
> > k_1^T & k_2^T & k_3^T \\
> > \end{bmatrix})
> > \cdot
> > \begin{bmatrix}
> > v_1 \\
> > v_2 \\
> > v_3 \\
> > \end{bmatrix}=
> > \mathrm{Softmax}(
> > \begin{bmatrix}
> > q\cdot k_1^T &q\cdot k_2^T &q\cdot k_3^T \\
> > \end{bmatrix})
> > \cdot
> > \begin{bmatrix}
> > v_1 \\
> > v_2 \\
> > v_3 \\
> > \end{bmatrix}=
> > \frac{\exp(q\cdot k_1^T)}{\gamma}\cdot v_1 + \frac{\exp(q\cdot k_2^T)}{\gamma}\cdot v_2 + \frac{\exp(q\cdot k_3^T)}{\gamma}\cdot v_3,
> > $$
> > where $\gamma=\sum_{i=1}^3{\exp(q\cdot k_i^T)}$ in Softmax.
> > If we change the relative order of the tokens, we can get
> > $$
> > \mathrm{Softmax}(q\cdot
> > \begin{bmatrix}
> > k_3^T & k_1^T & k_2^T \\
> > \end{bmatrix})
> > \cdot
> > \begin{bmatrix}
> > v_3 \\
> > v_1 \\
> > v_2 \\
> > \end{bmatrix}=
> > \mathrm{Softmax}(
> > \begin{bmatrix}
> > q\cdot k_3^T &q\cdot k_1^T &q\cdot k_2^T \\
> > \end{bmatrix})
> > \cdot
> > \begin{bmatrix}
> > v_3 \\
> > v_1 \\
> > v_2 \\
> > \end{bmatrix}=
> > \frac{\exp(q\cdot k_3^T)}{\gamma}\cdot v_3 + \frac{\exp(q\cdot k_1^T)}{\gamma}\cdot v_1 + \frac{\exp(q\cdot k_2^T)}{\gamma}\cdot v_2.
> > $$
> > As observed, the above two equations have the same result.
> >
> >
> > ---
> >
> > **Question 4:** Secure computation framework is still unclear.
> >
> > **Response 4:** Thanks for your thoughtful suggestions! We have further improve the overall framework and dataflow of MPCache in Figure 19 in our updated PDF, which is consistent with our description in Algorithm 3. We will add the framework in our final version.
> >
> > ---
> >
> > **Question 5:** If the similarity is calculated using the value cache, will the final performance be the same?
> >
> > **Response 5:** Not the same. The current LLM architectures are based on using the dot product of query and key cache to caculate the token similarity.
> > Also, in the attention computation, the relationship between query and key cache is further amplified by the Softmax function, while value cache is linearly multiplied.
> > Therefore, key cache has a much greater impact on the attention result than value cache.
> > That's why we choose to use key cache rather than value cache to calculate the similarity.
> >
> > Here, we also provide additional experimental results (F1 score) on LongChat-7B-32K and HotpotQA to support the point as shown in the following table.
> > As observed, using value cache to cacluate the similarity leads to significant accuracy degradation, especially for lower KV cache budgets.
> >
> > | KV Cache Budget | Use K to Measure Similarity | Use V to Measure Similarity
> > | --- | --- | --- |
> > | 3% | 29.55 | 13.32 |
> > | 6% | 30.27 | 19.51 |
> > | 12% | 30.00 | 25.67 |
> >
> > ---
> >
> > **Question 6:** Minor issues about the citation.
> >
> > **Response 6:** Thanks for your kind advice! We will carefully revise our citations in our revised version.

---

> > > ### Comment · Reviewer_782C · 2024-11-25
> > >
> > > Thanks to the authors for their efforts and detailed responses. The authors have corrected many issues such as the equality protocol in Algorithm 2 and improved the overall framework and unclear citations. I am willing to raise my rating to reflect these improvements. I encourage the authors to include these revisions, especially the secure computation protocols and framework in their final version. I have one question about the new clarification on TopK. Did you use the shuffle-based TopK in your plaintext experiments as well, such as those in Figure 9?

---

> > > > ### Author Response · Authors · 2024-11-26
> > > > **Thank you very much for your comments and support - Response to Reviewer 782C**
> > > >
> > > > Thank you very much for your thorough comments and support, which have greatly helped us improve our work! We will include all these revisions and carefully revise our writing in our final version.
> > > >
> > > > For your question, our plaintext experiments are conducted based on the PyTorch framework, therefore, we do not use shuffle-based top-k. Shuffle-based top-k does not impact the correctness, and our plaintext experiments are used to evaluate the effectiveness of our proposed algorithm.
> > > >
> > > > Thank you again for your attention and valuable suggestions.

---

> > > > ### Author Response · Authors · 2024-11-27
> > > > **Thanks for your comments and assistance - Response to Reviewer 782C**
> > > >
> > > > We would like to extend our heartfelt thanks once again for your valuable comments and assistance!
> > > > Also, we would like to know if you have any other questions or suggestions.

---

> > > > ### Author Response · Authors · 2024-12-02
> > > > **Follow-up on the Paper Rating for Paper 10561 - To Reviewer 782C**
> > > >
> > > > Dear Reviewer 782C,
> > > >
> > > > We are glad to hear your valuable suggestions and positive response in the rebuttal period. We noticed that you stated you are willing to raise your rating, but we find this is not yet reflected in the OpenReview rating. Since the deadline for the rebuttal period is approaching, we kindly request that you reflect your rating update in OpenReview by editing the rating on the original review.
> > > >
> > > > Thank you very much for your time and efforts in reviewing our work.

---

> ### Author Response · Authors · 2024-12-01
> **Follow-up on Rebuttal for Paper 10561 - To Reviewer 782C**
>
> Dear Reviewer 782C,
>
> As the deadline for the rebuttal period is approaching, we would like to know if your concerns about this paper have been resolved. If there are any other questions or concerns, please let us know.
>
> Thank you again for your time and effort in reviewing our work!

---

### Official Review · Reviewer_crCT · 2024-11-04

**Soundness:** 3
**Presentation:** 3
**Contribution:** 3
**Rating:** 6
**Confidence:** 3

**Summary:**

This work introduces a framework that makes KV-caching more "MPC-friendly", for use in the context of privacy-preserving LLM inference. Concretely, KV caching is a commonly used technique to reduce the amount of work when dealing with long input sentences in LLMs. In the MPC context, both the model and input sentence are private, and inference is carried out by an interactive protocol among the parties. Unfortunately, existing KV caching techniques still require several operations that, although cheap to execute "in the clear" (that is, when both the model and the input query are known locally), it is much harder to compute distributedly using an MPC protocol, since it uses operations that are difficult to carry out in MPC. The contribution of this paper is to design several methods that enable KV caching in MPC (that is, when both model and input query are secret). Concretely, the authors propose a mix of "static eviction", which eliminates some of the operations in a data-independent manner, followed by a more data-dependent (i.e. dynamic) eviction method, which compared to usual KV caching is carried out over less data and hence it's cheaper.

The authors present experiments that validate the effect of their techniques. They show that this leads to concrete gains in MPC while not affecting accuracy dramatically.

**Strengths:**

The problem is well motivated. This is part of the general umbrella of enabling private inference on LLMs using cryptographic techniques, and it is widely acknowledged that in general several modifications to the underlying ML algorithms would be required. This paper introduces a series of techniques that enable KV caching in MPC in a efficient manner. The results show improvement w.r.t. using standard KV caching techniques (which are not MPC-friendly). Detailed experiments are presented and implementation is included for reproducibility.

To the best of my knowledge, no explicit work in the direction of privacy-preserving LLM inference has studied explicitly (or even acknowledged) the use of KV caching for improving efficiency. I find the ideas presented in the paper to be interesting and novel, and the experiments to be sound.

**Weaknesses:**

I am doubtful of the ability of these techniques to generalize to other models. The proposed method involves dropping a huge percentage of the intermediate values, since the authors find out experimentally that they contribute little to the attention heads. However, this study, which is explored deeply in the paper, is highly tailored to a specific model and datasets. This is of course acceptable and it's the usual way to study the viability of new ML techniques. However, in this specific setting the situation is more critical since an actual deployment in a real scenario would likely involve a model that is intended to be private. I don't see a clear way to deploy the techniques involved presented here without performing extensive experimentation that would require knowledge of the model (or at least leaking information about it beyond the architecture).

Overall, to summarize my concerns: I feel the techniques presented here work very well for the model evaluated, but I have no particular reason to believe this would be useful in more realistic settings where both the model and the data are private. Perhaps a scenario where a single party knows the model and also has the data to identify the best "hyper-parameters" for static-eviction would work, but this is not the most general scenario and this is not explored in the paper.

**Questions:**

What's the exact distributed protocol used for performing the dynamic eviction? My understanding is that this is performed by the parties obliviously, and I understood the description of the algorithm. However, this involves sorting and other complex cryptographic operations. I am unsure the authors describe the exact protocols used for this part.

Can you comment on your thoughts regarding how reasonably your method generalizes, and how it can be used in oblivious settings where the only known aspect of a model is its architecture? (and in particular, it is not possible to benchmark the accuracy of the method).

Suggestion:

* Some citations are inaccurate or missing, particularly when it comes to MPC. The 3PC protocol from Section B.3 is from Furukawa et al (CCS'17). I don't understand why Bumblebee claimed to be the SOTA for 2PC, especially when there are PCGs for non-interactive preprocessing that avoid computational assumptions in the online phase. There is also the Sigma work (SIGMA: Secure GPT Inference with Function Secret Sharing) that does 3PC as well (2PC + preprocessing), and it is not mentioned. In general the work seems to have a bias towards some specific prior works.

---

> ### Author Response · Authors · 2024-11-20
> **Response to Reviewer crCT**
>
> We appreciate Reviewer crCT's constructive feedback, which helps us improve our work! Below we answer each of your comments and questions:
>
> ---
>
> **Question 1:** Ability of these techniques to generalize to other models. How the method would be useful in more realistic settings where both the model and the data are private.
>
> **Response 1:**
> We believe our proposed static-dynamic eviction algorithm has good generalization capability. This is because of two reasons: 1) redundant tokens that do not impact the decoding of any downstream tokens generally exists in different input sentences, which is observed in many previous works. These tokens can thus be statically evicted without impacting the LLM performance. In our experimentals, we use the static eviction algorithm to prune ~50\% tokens, which is very different from [1,2] that only rely on static eviction and only prune once to the specific ratio; 2) the effectiveness and generalization ability of dynamic eviction originates from the Softmax attention. As the Softmax function normalizes the summation of the attention scores to 1, it limits the maximum number of tokens that impact each LLM decoding step. We observe the sum of attention scores of the top-100 ranked tokens is usually larger than 0.95, indicating majority of the impact comes from these top-100 ranked tokens. This observation makes us confident to use a high eviction ratio during the dynamic eviction phase without significant accuracy degradation.
>
> In practice, we believe the static and dynamic eviction ratio should be determined based on two aspects: 1) the expected sequence length and 2) the latency that can be tolerated/afforded by the server and the client. Our framework proposes a novel algorithm with configurable hyper-parameters to enable exploring the Pareto front of the LLM performance and efficiency. The framework has been validated across different benchmarks and different LLMs, which demonstrates its value and potential for practical usecases.
>
> In our revised version, we will provide more discussions on the generalization capability.
>
> ---
>
> **Question 2:** Exact protocols used for performing the dynamic eviction.
>
> **Response 2:**
> The protocols used for dynamic eviction can be divided into the following three parts: 1) Similarity approximation. As shown in Equation (6), we invoke the **matrix multiplication protocol** to measure the relationship between the query token and previous tokens. 2) Top-k selection. We then invoke **top-k protocol** to obtain the indices of important tokens. The protocol is the same as that in CipherGPT and we optimize the protocol with parallelization for fast head-wise token selection. 3) Token gathering. This protocol consists of two sub-protocols: a) **comparison protocol** to securely convert integer indices into one-hot vectors; and b) **matrix multiplication protocol** of the KV cache and one-hot vectors to obtain the selected tokens. We describe the details in Appendix B.4.
>
> In this work, our focus is on the algorithm optimization for efficient inference rather than protocol optimization, so we directly follow the protocols proposed by 3PC PUMA and 2PC BumbleBee, which are implemented in the SecretFlow framework. We will provide detailed protocol descriptions more clearly in our revised version.
>
> ---
>
> **Question 3:** Some citations are inaccurate or missing.
>
> **Response 3:**
> Thanks for your kind suggestion! We have revised the incorrect citations and added missing citations in our updataed version. We will further check the citations in our final version.
>
> ---
>
> **References:**
>
> [1] Li, Yuhong, et al. "Snapkv: Llm knows what you are looking for before generation." arXiv preprint arXiv:2404.14469 (2024).
>
> [2] Cai, Zefan, et al. "Pyramidkv: Dynamic kv cache compression based on pyramidal information funneling." arXiv preprint arXiv:2406.02069 (2024).

---

> > ### Comment · Reviewer_crCT · 2024-11-26
> >
> > Thanks for your response. Your clarifications make sense. For the exact protocols used for dynamic eviction I would suggest more clarity in the main body of the paper.

---

### Meta-Review · Area_Chair_2exy · 2024-12-21

**Metareview:**

This submission introduces an MPC-friendly implementation of KV-caching in the context of secure and efficient LLM inference.
The main concern about this paper is that the algorithm is not clearly specified in the write-up. While some inconsistencies that were raised by the reviewers were addressed by the authors during the discussion, the paper would benefit from a further improvement in the presentation, including the details of the algorithm and the threat model, before being ready for publication.

**Additional Comments On Reviewer Discussion:**

The reviewers raised different concerns about this submission including:

1) Missing prior related work
2) Unclear implementation
3) No security proofs
4) Setting of MPCache hyperparameters
5) Unclear presentation
6) Motivation

While the reviewers have attempted to answer these in the discussion, concerns remained about 2) and 5).

---

### Decision · Program_Chairs · 2025-01-22

Reject